# Evidence of, and a Proposed Explanation for, Bimodal Transport States in Alluvial Rivers

Kieran B.J. Dunne[1] and Douglas J. Jerolmack[1]

[1]Department of Earth and Environmental Science, University of Pennsylvania, Philadelphia, Pennsylvania, USA.

*Correspondence to:* Kieran Dunne (kdunne@sas.upenn.edu)

**Abstract.** Gravel-bedded rivers organize their bankfull channel geometry and grain size such that shear stress is close to the threshold of motion. Sand-bedded rivers on the other hand typically maintain bankfull fluid stresses far in excess of threshold, a condition for which there is no satisfactory understanding. A fundamental question arises: Are bed-load (gravel-bedded) and suspension (sand-bedded) rivers two distinct equilibrium states, or do alluvial rivers exhibit a continuum of transport regimes as some have recently suggested? We address this question in two ways: (1) re-analysis of global channel geometry datasets, with consideration of the dependence of critical shear stress upon site-specific characteristics (e.g. slope and grain size); and (2) examination of a longitudinal river profile as it transits from gravel to sand-bedded. Data reveal that the transport state of alluvial river-bed sediments is bimodal, showing either near-threshold or suspension conditions, and that these regimes correspond to the respective bimodal peaks of gravel and sand that comprise natural river-bed sediments. Sand readily forms near-threshold channels in the laboratory and some field settings, however, indicating that another factor, such as bank cohesion, must be responsible for maintaining suspension channels. We hypothesize that alluvial rivers adjust their geometry to the threshold-limiting bed and bank material — which for gravel-bedded rivers is gravel, but for sand-bedded rivers is mud (if present) — and present tentative evidence for this idea.

## 1  Introduction

Almost 100 years ago, Lacey (1930) proposed an empirical relationship relating the width of an alluvial river to its water discharge. Leopold and Maddock (1953) built upon this to derive the hydraulic scaling relations for bankfull channel geometry of alluvial rivers. Decades of research since have added geographic (Parker et al., 2007; Richards, 1987) and morphologic (e.g., braided vs. meandering, Gaurav et al. (2015); Métivier et al. (2016)) variety to data compilations, and recognized the importance of vegetation and geologic controls that were not originally considered (e.g., Huang and Nanson (1998); Schwendel et al. (2015); Kleinhans et al. (2015); Nanson and Young (1981); Ferguson (1987)). Yet the original findings are robust: bankfull width ($W_{bf}$), bankfull depth ($H_{bf}$) and slope ($S$) scale as power-law functions of bankfull discharge ($Q_{bf}$) with little variation in the exponents (Parker et al., 2007), suggesting a simple and common organizing principle for alluvial rivers. Cast in dimensionless form following Métivier et al. (2016) and Andrews (1984), with $Q_* = Q_{bf}/\sqrt{RgD_{50}^5}$ where $D_{50}$ is the

river-bed median grain size, $R$ is the particle submerged specific gravity, and $g$ is gravity, the often-called "regime equations" read:

$$W_{bf}/D_{50} = \alpha_W Q_*^{\beta_W}$$
$$H_{bf}/D_{50} = \alpha_H Q_*^{\beta_H}$$
$$S = \alpha_S Q_*^{\beta_S} \tag{1}$$

where $\alpha$ and $\beta$ are dimensionless parameters. The theoretical underpinning of the regime equations (1) is both well known and
elusive; it is the equilibrium channel geometry problem (Leopold and Maddock, 1953). Considering fluid mass conservation in a rectangular channel:

$$Q_{bf} = u_{bf} H_{bf} W_{bf}, \tag{2}$$

and friction via a Chezy-type relation:

$$u_{bf} = \sqrt{g H_{bf} S}/C_f, \tag{3}$$

where $u_{bf}$ and $C_f$ are average bankfull flow velocity and friction factor, respectively, we obtain two relations among the governing hydraulic variables. If $Q_{bf}$, $D_{50}$ and $C_f$ are specified (as is typical), one still requires an additional relation among the parameters to close the set of equations and derive equation 1 (Métivier et al., 2017).

   "Regime theory" is the application of these agreed upon relationships with one additional threshold channel based-assumption to allow for closure. There are three dominant branches of regime theory, each with their own form of a threshold channel clo-
sure assumption that separate regime theory into three distinct schools of thought: 1) assume that river are canals, and thus threshold channels; 2) assume that the transport regime is purely bedload and solve the 2-D flow field to balance fluid shear stress and particle weight at the edge of the channel, while simultaneously allowing for transport at the center; 3) assume that the river undergoes an optimization process that maximizes friction in order to reduce fluid shear stress, ultimately resulting in a threshold channel.

The first school of thought is based upon work done to calculate the shape of a stable canal for which the bed material is at the threshold of motion (Glover and Florey, 1951). This work has been extended to natural rivers by Henderson (1961), and offers an explanation for observations of alluvial river width relating to the water discharge (Henderson, 1961; Andrews, 1984; Métivier et al., 2017). This line of thinking links well in with the second branch of a regime theory which as was established by Parker (1978a). Parker (1978a) solved the 2-D stress field to show that, for a pure bedload river, the channel is
at the threshold of motion for the material at the banks and slightly above the threshold of motion in the center, allowing for the river to transport sediment, while at the same time maintaining a stable and consistent width. This model is supported by both global compilations of data and case studies of individual rivers that demonstrate that bedload-dominanted gravel-bedded

rivers are slightly offset from a threshold channel (Phillips and Jerolmack, 2016; Gaurav et al., 2015; Métivier et al., 2016). Several studies have presented evidence that sediment supply and bank vegetation may drive gravel channels further above threshold (Pfeiffer et al., 2017; Millar and Quick, 1998). Values for Shields stress in gravel-bed rivers reported for a wide range of environments, however, rarely exceed 2-3 times critical.

Parallel to this grain size-dependent channel geometry is the concept of optimization which assumes that rivers seek a threshold channel condition by maximizing the flow resistance within the channel to minimize the fluid shear stress (Eaton et al., 2004; Eaton and Church, 2007). The rational regime theory put forward by Eaton attempts to infer the importance of bank strength given deviations away from the threshold condition that is posited by optimality theory (Eaton et al., 2004; Eaton and Church, 2007). However, these relationships are predominantly calibrated on coarse-grained rivers where research has

shown that the influence of cohesive mud is a minor control on the erodibility compared to the weight of the gravel (Kothyari and Jain, 2008). What distinguishes our work from this work is that we extend the concept of Parker's threshold channel model into the space occupied by fine-grained rivers by the suggestion that river channel geometries, and their subsequent sediment transport state are either controlled by the erodibility of their beds or their banks. This paper shows the transition from rivers that can be explained entirely by Parker's theory (i.e. channel beds and banks composed of uniform material transported entirely in

bedload) to channels that cannot. For natural rivers, this transition most frequently occurs at the transition from a gravel-bedded to a sand-bedded condition. This transition coincides with the point at which bed material becomes small enough such that the cohesion of channel banks should become important. What we show is that the sediment transport state is bimodal because grain size is bimodal; the coarser gravel mode is more difficult to entrain than any cohesive bank material, while the finer sand mode is easier to entrain than any cohesive bank material (if present).

As nicely summarized in a recent series of papers (Métivier et al., 2016; Gaurav et al., 2015; Métivier et al., 2017), a useful starting point for the equilibrium channel geometry problem is to consider what we call here the "ground state" in which no sediment transport occurs. In this situation, which may be achieved in a laboratory experiment with a constant $Q_{bf}$ and no sediment feed, the river organizes such that the boundary shear stress everywhere along the channel cross section is at the threshold of motion (Métivier et al., 2017). Accordingly, the local and width-averaged bankfull Shields stress should be at the

critical value, $\tau_{*bf} = \tau_{*c}$, and may be estimated assuming normal flow as:

$$\tau_{*bf} = \frac{H_{bf}S}{RD_{50}} \qquad\qquad (4)$$

where $R = 1.65$ is the assumed relative submerged grain density. Setting equation 4 equal to $\tau_{*c}$ provides the necessary closure to determine channel geometry, as first illustrated by Lacey (Lacey, 1930) who solved for the shape of a canal. Of course, natural rivers are not canals; they transport sediment, which requires that their formative Shields stress be larger than critical.

Compilations of channel geometry and Shields stress, using global datasets, reveal that alluvial rivers naturally break out into two classes: gravel-bed rivers ($D_{50} > 10mm$) in which $1 \leqslant \tau_{*bf}/\tau_{*c} \leqslant 2$, and sand-bed rivers ($D_{50} < 1mm$) with $\tau_{*bf}/\tau_{*c} >> 1$. The scaling exponents (equation 1) for both classes are similar and in reasonable agreement with predictions from "Lacey's

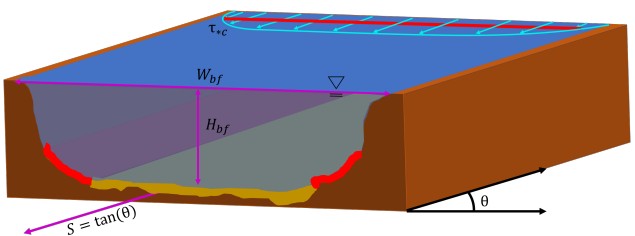

**Figure 1.** Schematic cross-section of a sand-bedded, alluvial river with different bed vs. bank material, under bankfull flood conditions. Here $W_bf$, $H_bf$, and $S$ are bankfull width, bankfull depth and channel gradient at the cross-section, respectively. Cyan lines at surface illustrate horizontal stress profile across the channel. Red lines along channel bottom indicate toe of the river bank — i.e., the intersection of bed and bank material. Red line intersecting the cyan velocity profile indicates the threshold stress of the threshold-limiting material, illustrating that the bank toe is at threshold while Shields stress in the channel center is slightly in excess of threshold.

law", however, the coefficients are different from each other and the threshold channel (Métivier et al., 2017, 2016; Gaurav et al., 2015).

Parker (1978a) provided the first generalization of the threshold channel theory to gravel-bed rivers, which transport sediment as bed load. He recognized that stable river banks are incompatible with transport; the transverse slope drives a net flux away
from the bank, leading to erosion and channel widening. The solution to the so-called "stable-channel paradox" (Parker, 1978b) lies in the lateral (cross-stream) gradient in bed stress — flow velocity and depth increase with distance away from the bank. An equilibrium channel may therefore be constructed that is marginally above threshold in the center but at threshold on the banks. Parker (1978a) predicted $\tau_{*bf}/\tau_{*c} \approx 1.2$ for equilibrium bed-load rivers, in agreement with observations of natural gravel-bed rivers (Paola et al., 1992; Parker et al., 1998; Dade and Friend, 1998; Parker et al., 2007; Phillips and Jerolmack,
2016) and laboratory experiments (Ikeda et al., 1988; Pitlick et al., 2013; Reitz et al., 2014). In terms of the regime equations 1, gravel-bed rivers thus follow expectations from the threshold theory but with a slight offset due to their higher bankfull Shields stress (Métivier et al., 2017). Parker (1978b) also realized that sandy (suspension) rivers cannot behave in a similar manner, in that boundary stresses even at the channel margins would be above threshold leading to erosion. In order to counter slope-driven bank erosion, Parker (1978b) and subsequent researchers (Ikeda and Nishimura, 1985; Ikeda et al., 1988; Wilkerson
and Parker, 2010) proposed that lateral diffusion of suspended sediment outward from the channel center could compensate for inward bed-load sediment transport from the banks. While physically reasonable, suspension channel theories have not provided a satisfactory description of sandy river channel geometry. At present there is no accepted model for the equilibrium geometry of river channels far above threshold.

In the absence of a theory, subsequent research has focused on examining trends drawn from compilations of data on channel
hydraulic geometry and bankfull discharge. Examination of gravel-sand transitions along downstream river profiles indicates that the mode of bed-material transport may switch abruptly from near-threshold (gravel-bedded) to suspension (sand-bedded) (Miller et al., 2014; Venditti et al., 2015, 2010; Singer, 2010, 2008; Blom et al., 2017), and hydraulic considerations have suggested that susceptibility to suspension increases rapidly as grain size decreases across the gravel to sand range (Lamb

and Venditti, 2016). On the other hand, recent compilations of global data sets have been used to suggest that rivers exhibit a continuum of transport states — from near threshold through to full suspension — and that bankfull Shields stress varies smoothly with grain size, slope and particle Reynolds number (Parker et al., 2007; Wilkerson and Parker, 2010; Li et al., 2015; Trampush et al., 2014)). Importantly, this new presentation of the data suggests that there is no range in phase space where rivers cluster near the ground state of a constant threshold Shields stress (Fig. 4). Phillips and Jerolmack (2016) found, however, that gravel-bed rivers do indeed cluster close to the threshold of motion — if the dependence of threshold upon site-specific characteristics (e.g. slope or grain size (Lamb et al., 2008; van Rijn, 2016)) is explicitly accounted for. Moreover, while previous data compilations found that bankfull Shields stress increases systematically with decreasing grain size (Li et al., 2015; Trampush et al., 2014), one may readily find data that contradict this trend. Channels formed by seepage erosion in sand (Devauchelle et al., 2011; Marra et al., 2015) are observed to transport sand as bedload and, like gravel bedload rivers, cluster approximately at the threshold of motion. Similarly, sand-bedded rivers in laboratory experiments also form near-threshold channels (Reitz et al., 2014; Métivier et al., 2016; Federici and Paola, 2003).

We are left with three questions that will be considered in this paper. First, how do rivers transition from near-threshold to suspension states? Second, is the near-threshold channel an attractor, or merely a limiting state? And third, how do suspension rivers maintain an equilibrium channel geometry? We address these questions by re-analysis of existing data. We revisit the global data compilations of Li et al. (2015) and Trampush et al. (2014), and argue that natural rivers appear to exhibit bi-modal transport states corresponding to near threshold (order 1 multiplier of threshold) and far-above threshold (order 10-100 multiplier of threshold). We also show that this bi-modal behavior is exhibited within a single river profile transiting the gravel to sand transition. These results lend credence to the hypothesis first put forward by Lane (1937) and then Schumm (1960): Alluvial rivers adjust their geometry to the threshold-limiting bed and bank material. It follows that sand-bed rivers may be suspension channels if their banks are composed of more resistant material (Church, 2006), e.g., cohesive sediment and/or vegetation. Gravel rivers, on the other hand, should be less sensitive to bank composition due to the relatively high threshold stress for entrainment of coarse grains (Schumm, 1960).

## 2 Data Sources

The large, global datasets utilized in this paper are identical those used by Trampush et al. (2014) and Li et al. (2015). They were subsequently combined with a longitudinal profile from the Sacramento River (Singer, 2010), river channel cross sections on the Kosi Megafan (Gaurav et al., 2015), and channels formed by seepage erosion in the Apalachicola ravines in Florida (Devauchelle et al., 2011) and in a laboratory (Reitz et al., 2014). This combination of data allows for the following comparisons between localized examples and global trends in river channel characteristics: 1) how changes in hydraulic geometry and sediment transport regime that a single river experiences across a gravel-sand transition compare to exhibited global trends in hydraulic geometry and Shields stress; and 2) how rivers that originate in sandy substrates with little cohesion compare in terms of hydraulic geometry and sediment transport regime to channels with gravel beds. All data used in this analysis (Li

et al., 2015; Trampush et al., 2014; Singer, 2010; Gaurav et al., 2015; Devauchelle et al., 2011; Reitz et al., 2014) are available as supplementary material and include bankfull estimates of width, depth, slope, grain size, and discharge.

Our re-analysis requires that we estimate the critical Shields stress for incipient motion, $\tau_c^*$, for each data point. Determination of $\tau_c^*$ is a notorious problem (Buffington and Montgomery, 1997; Mueller et al., 2005; Lamb et al., 2008; van Rijn, 2016) and, despite the best efforts of researchers, no theory can reliably predict values for the field. Nevertheless, there is strong field and laboratory evidence that $\tau_c^*$ varies with site-dependent characteristics, such as slope (Mueller et al., 2005; Lamb et al., 2008; Phillips and Jerolmack, 2014, 2016) and grain size (Shields, 1936; van Rijn, 2016). In this study we use and compare the empirically-determined slope-dependent relation of Lamb et al. (2008):

$$\tau_c^* = 0.15 S^{0.25}, \tag{5}$$

to the Shields-curve fit of van Rijn (2016):

$$\tau_{*c} = \frac{0.3}{1 + D_*} + 0.055 \left(1 - e^{-0.02 D_*}\right) \tag{6}$$

where $D* = (Rg)^{1/3} D_{50}/\nu^{2/3}$ is dimensionless grain size and $\nu$ is kinematic viscosity. We note that our findings change little if we use the linear slope-dependent relation of Mueller and Pitlick (2005) instead of equation 5.

## 3 Hydraulic Geometry Scaling Revisited

We first examine hydraulic geometry scaling as suggested by the regime equations 1. For comparison, we also compute the expectations for a threshold channel following Métivier et al. (Métivier et al., 2016; Gaurav et al., 2015):

$$
\begin{aligned}
\frac{W_{bf}}{D_{50}} &= \left[ \frac{\pi}{\sqrt{\mu}} \left(\tau_{*c}\right)^{-1/4} \sqrt{\frac{3 C_f}{2^{3/2} K[1/2]}} \right] Q_*^{1/2}; \\
\frac{H_{bf}}{D_{50}} &= \left[ \frac{\sqrt{\mu}}{\pi} \left(\tau_{*c}\right)^{-1/4} \sqrt{\frac{3\sqrt{2} C_f}{K[1/2]}} \right] Q_*^{1/2}; \\
S &= \left[ \left(\sqrt{\mu} \tau_{*c}\right)^{5/4} \sqrt{\frac{2^{3/2} K[1/2]}{3 C_f}} \right] Q_*^{-1/2}.
\end{aligned}
\tag{7}
$$

For simplicity, we choose values for the following coefficients to be identical to those reported in Métivier et al. (2016): Chezy friction factor $C_f \approx 0.1$, Coulomb friction coefficient $\mu \approx 0.7$, and $K[1/2] \approx 1.85$. These values could be manipulated to enhance their fit to data if desired, but this exercise is not performed here. We treat $\tau_{*c}$ in two ways: (1) assuming a constant critical Shields stress with a representative gravel-bed river value $\tau_{*c} = 0.03$ as in Métivier et al. (2016); and (2) using the slope and grain size dependent critical values from equations 5 and 6, respectively.

To first order, gravel- and sand-bedded rivers could be described by a single continuous power-law relation for dimensionless channel width $W_{bf}/D_{50}$ as a function of $Q_*$. A second-order feature is present, however, in the high $Q_*$ limit; a subset of

sand-bed streams show an upward offset from the general trend (Fig. 2). Dimensionless channel depth $H_{bf}/D_{50}$ shows similar behavior, except that the high-$Q_*$ sandy streams show a downward rather than upward offset. In general, gravel-bed rivers are close to threshold predictions while sand-bed streams depart more significantly, similar to earlier findings by Métivier et al. (Métivier et al., 2016; Gaurav et al., 2015). Both constant and slope-dependent threshold channel predictions capture the general trends, but predict a systematically steeper scaling exponent than is exhibited by the data. Slope has a behavior that is distinctly different from width and depth; sand-bedded rivers in general display a large offset from the gravel-bedded river trend, and a correspondingly large offset from threshold channel predictions (Fig. 2). Slope exhibits more scatter than channel geometry, a common pattern in river data compilations that likely reflects the long timescale associated with slope adjustment (Métivier et al., 2016; Gaurav et al., 2015). Note that, for all variables, the sandy seepage erosion channels in Florida generally plot with the gravel-bedded river data showing that sand-bedded rivers do not necessarily behave differently from gravel-bedded ones.

One interesting finding is that the product of dimensionless width and depth, i.e., dimensionless channel cross-sectional area, shows the tightest relation to $Q_*$ and no offset between gravel- and sand-bed channels. This is noteworthy considering that width and depth plots show considerable scatter, so one might expect that their product would exhibit more scatter if the variability was due to random noise or error. This suggests that rivers systematically increase their cross-sectional area $A$ to accommodate increasing discharge — regardless of grain size and transport stage; in other words, $A$ is primarily controlled by hydraulics alone (indeed flow resistance, and hence flow velocity $u_{bf}$, is approximately independent of channel aspect ratio for values $W_{bf}/H_{bf} > 10$ (Guo and Julien, 2005) that are typical of natural rivers). How changes in $A$ are partitioned into width vs. depth, however, may depend on bed/bank substrate and sediment transport conditions.

## 4  Bimodality in the Transport States of Global Datasets

As the name implies, hydraulic geometry scaling does not consider the transport state of sediment within channels. A simple way to do so is consideration of the bankfull Shields stress $\tau_{*bf}$. Earlier global compilations of river data suggested that transport states were bimodal, with gravel-bed rivers clustering around a Shields stress close to the critical value ($\tau_{*bf} \sim 10^{-2}$) and sand-bed rivers clustering around a much higher value ($\tau_{*bf} \sim 10^0$) (Paola et al., 1992; Parker et al., 1998; Dade and Friend, 1998). Indeed, we see compelling evidence for this bimodality across a range of slopes and grain sizes in our global compilation (Fig. 3). There are clear deviations from this trend, however; the sandy Florida seepage channels (Devauchelle et al., 2011) and sandy laboratory experimental rivers of Reitz et al. (2014) both plot in the range of phase space otherwise occupied by gravel-bed rivers. What these channels have in common is that they are small, sand-bedded rivers with bank material that is similar in composition to the bed (i.e., sandy).

The case for a continuum of transport states was made more recently by Li et al. (2015), who showed that $\tau_{*bf}$ is inversely proportional to dimensionless grain size $D_*$ and scales with roughly the square root of $S$. They presented a similarity collapse for the data with a best-fit relation $\tau_{*bf}/S^{0.53} = 1220 D_*^{-1}$, and a similar result was found by Trampush et al. (2014). Li et al. (2015) concluded that the notion of a constant formative Shields stress for either gravel- or sand-bedded channels was not

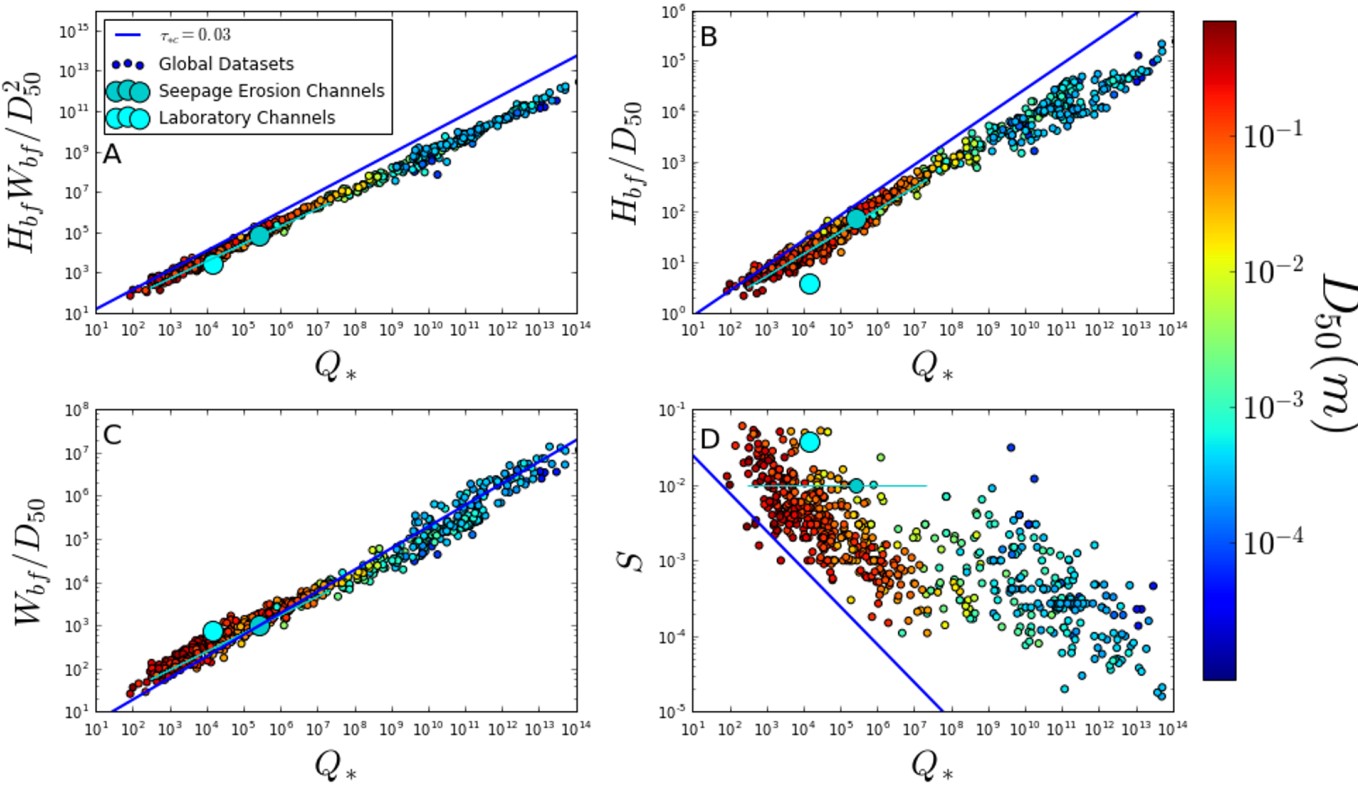

**Figure 2.** Dimensionless hydraulic geometry scaling for rivers in the global data set. (A) Cross-section area shows a tight relation with discharge across the entire range of data. (B) Depth and (C) width follow similar first-order trends for gravel vs. sand bed rivers, but with some offset between these groups. (D) Slope separates sand and gravel rivers. Blue line shows exceptions from the threshold equations (7) assuming a constant reference Shields stress for simplicity. We note that the fit does not improve if grain-size or slope dependent threshold predictions are used instead. Larger points illustrates the mean of multiple measurements taken along a single longitudinal profile. Cyan error bars represent the range of data, and are used because the original study reported only one value for slope and for grain size for all cross sections (Devauchelle et al., 2011).

supported by the data. We reproduce the figure of Li et al. (2015) here, where the addition of new data (discussed in the previous section) generally supports the similarity collapse (Fig. 4). The sandy Florida seepage channels and experimental rivers, however, fall off of this trend.

By assessing transport stage using bankfull Shields stress alone, previous authors either explicitly (Parker et al., 1998, 2007; Wilkerson and Parker, 2010) or implicitly (Li et al., 2015; Trampush et al., 2014) assumed that the critical Shields stress was constant. A recent study by Phillips and Jerolmack (2016), however, showed that, when site-specific variations in $\tau_{*c}$ are taken into account, gravel-bedded rivers exhibit a bankfull Shields stress that is close to the threshold value. We consider transport stage as $\tau_{*bf}/\tau_{*c}$. To test for the influence of variations in $\tau_{*c}$, we examine the distributions of Shields stress and transport stage

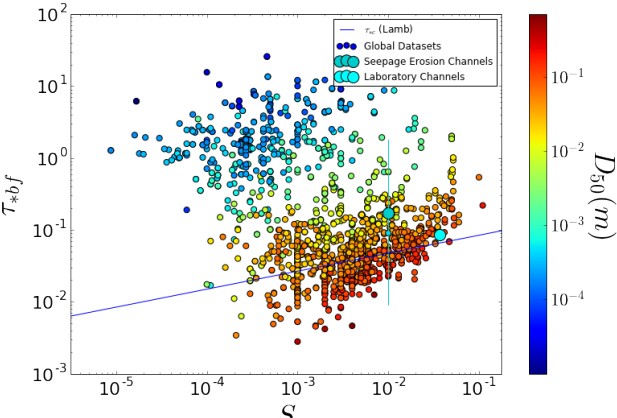

**Figure 3.** Bankfull Shields stress as a function of stream gradient. Coarse-grained rivers exhibit low Shields stresses with a moderate dependence on slope, that roughly follows but is offset from the slope-dependent relation of Lamb et al. (2008) for critical Shields stress (solid line). Fine-grained rivers cluster well in excess of the threshold of motion. River channels originating in sandy substrates found in the natural (Devauchelle et al., 2011) or laboratory (Reitz et al., 2014) environments are shown to be in the Shields stress space typically populated by gravel-bedded rivers.

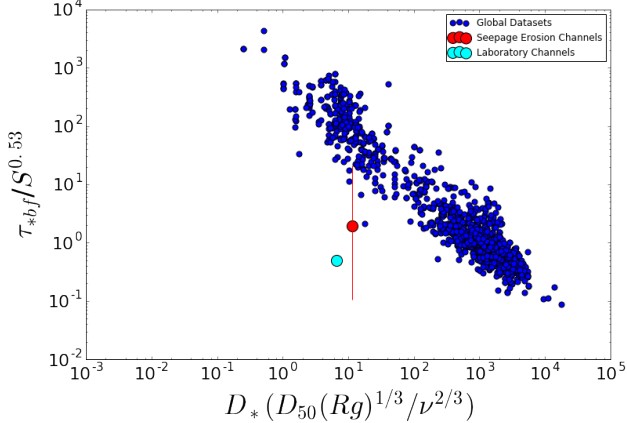

**Figure 4.** A re-creation of the diagram from Li et al. (2015) that makes the case for a continuum of sediment transport regimes. Additional data have been added to the diagram from an additional global dataset (Trampush et al., 2014), and various longitudinal profiles (Singer, 2010; Gaurav et al., 2015; Devauchelle et al., 2011). Clear deviations from the trend are demonstrated by river channels formed by seepage erosion in sand (with mean and error bars same as in Fig. 3), and channels formed in sand in laboratory experiments (Reitz et al., 2014) that are represented by the larger red and cyan points, respectively.

where for the latter $\tau_{*c}$ is estimated from either slope or grain size following equations 5 and 6. The Shields stress distribution is bimodal (Fig. 3). This bimodality becomes slightly more evident in the distributions of transport stage, though there is little difference between the results using the two different estimates for $\tau_{*c}$ (Fig. 5 B, C). The bimodality in Shields stress and

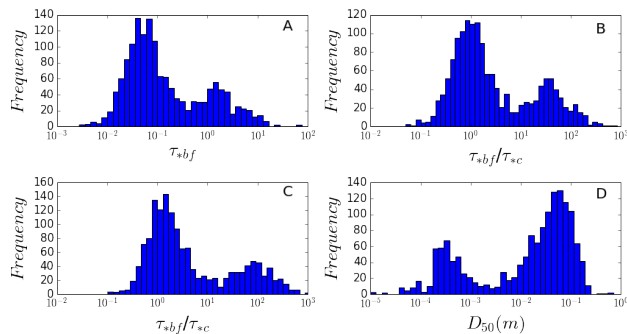

**Figure 5.** Distributions of (A) Shields stress, (B) Transport stage estimated using grain size (van Rijn, 2016), (C) Transport stage estimated using slope (Lamb et al., 2008), and (D) median river-bed grain size. All distributions are bimodal, with near-threshold gravel rivers and far-above threshold sandy rivers.

transport stage is mirrored by a comparable bimodality in river-bed grain size (Fig. 5 D). These findings revive the possibility of a constant transport-stage condition that is either close to or far above threshold, but also show that river-bed grain size is insufficient to predict transport stage as threshold sand-bed rivers may readily be found.

## 5    Bimodality in Transport Stage along a Longitudinal River Profile

The global dataset reveals an apparent dichotomy of transport states that generally (but not always) correspond to sand- or gravel-bedded rivers, but the nature of this dichotomy may be partially obscured by confounding variables among disparate river systems that are not accounted for. A useful complementary approach is to examine the longitudinal profile of a single river as it transits from gravel- to sand-bedded. We utilize data collected by Singer (2010) in his study of the gravel-sand transition of the Sacramento River. We can see that Shields stress is slightly in excess of critical for the gravel-bed portion of

the river, and far above critical for the sandy portion (Figure 6). In the gravel-sand transition we observe a flickering between these two distinct states, that is indicative of patchiness of bed materials (Singer, 2010). The fluid shear stress gradually declines downstream (Fig. 6 A), and width decreases across the gravel-sand transition but only modestly (Singer, 2010). Bed-sediment size changes abruptly, showing that the large variations in transport stage are overwhelmingly driven by the grain-size pattern (Figure 6 B). In summary, the Sacramento River shows the same bimodal behavior as the global dataset, in terms of transport

stage and grain size. Other factors such as slope or hydraulic geometry do not show this bimodality.

## 6    Discussion

It has long been suggested that bank composition influences the hydraulic geometry of rivers, under the premise that effective bank cohesion (silt/clay or vegetation) increases the threshold shear stress which leads to narrower and deeper channels. The evidence from gravel-bed rivers is that the cohesive effect is significant but modest; bank strength changes of up to two orders

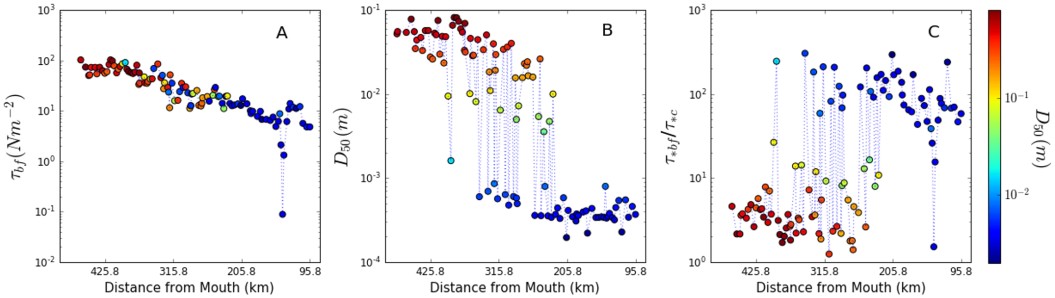

**Figure 6.** Longitudinal profile data from the Sacramento River up to approximately 500km upstream of the river mouth (Singer, 2010). (A) Fluid shear stress decreases gradually and continuously across the gravel-sand transition. (B) Grain size behavior downstream is bimodal, changing rapidly from gravel to sand. (C) Shields stress shows abrupt transition from near-threshold to far above threshold across the gravel-sand transition.

of magnitude correspond to differences in width of 2-3 times (e.g., Andrews (1984); Millar and Quick (1993); Millar and Quick (1998); Huang and Warner (1995); Huang and Nanson (1998)). Though there are far fewer studies on sand-bed alluvial rivers, the limited data indicate that the influence of bank cohesion may be larger in these systems (Kleinhans et al., 2015, 2014). The classic study by Fisk (1944) of the Mississippi River showed major narrowing and deepening as the river moved from sandy to clay-rich alluvium, while Schumm (1960; 1963) demonstrated that channel aspect ratio ($W_{bf}/H_{bf}$) was inversely proportional to the percent silt-clay (a proxy for cohesion) in the bed and banks of sand-bed rivers. Interestingly, he found no correlation between aspect ratio and percent silt-clay for gravel-bed rivers (Schumm, 1960). More recent studies on deltaic and tidal channels have also shown that bank strength strongly influences channel geometry (Kleinhans et al., 2009; Edmonds and Slingerland, 2010).

Lane (1937) and Schumm (1960) argued that channels initially cutting into alluvium should widen "until the resistance of the banks to scour prevents it" (Schumm, 1960). We rephrase this idea to posit a more specific hypothesis: Alluvial rivers, on average, organize their geometry such that the fluid shear stress at the toe of the bank is at the threshold of motion for the bankfull flow (Fig. 1). We consider the bank toe because (1) this is the zone of maximum fluid stress on the bank, and (2) bank-toe erosion is required to undermine upper bank materials. While slumping and block failures may strongly influence the rate of bank erosion, with important consequences for river dynamics such as meandering (Parker et al., 2011), these processes likely have little effect on average channel size. For rivers in which the bed and the bank toe are made of the same material — such as laboratory experiments, and some natural channels in non-cohesive sediments — we expect to recover the near-threshold "bed-load river" channel predicted by the Parker (1978a) model. For the more common case of rivers having a bank-toe composition that is different from the bed — typically cohesive and/or vegetated banks — we propose that alluvial rivers adjust their geometry to the threshold-limiting material. Thus, in order to maintain a "suspension river" like most natural sand-bed channels, the banks must be composed of cohesive sediment with a significantly higher entrainment threshold than

the bed material. Indeed, Church (2006), noted that sand-bed channels often have silt-clay banks that experience little to no deformation, while channel-bed sands are completely suspended.

Unfortunately, reported measurements of hydraulic channel geometry rarely include information about bank materials. To test the threshold-limiting idea indirectly, we consider the relative mobility of bed and bank materials as a function of grain size. We do not consider vegetation explicitly; however, we note that numerous studies have analyzed the effects of vegetation on erosion thresholds (Micheli and Kirchner, 2002; Abernethy and Rutherfurd, 2001). It is important to point out that Shields stress is not the relevant parameter for cohesive materials, where particle weight does not adequately describe resistance to motion. Dimensional fluid threshold stress is usually reported in studies involving cohesive sediment. Considering non-cohesive materials and neglecting slope effects, the threshold fluid stress determined from the Shields curve increases monotonically with increasing grain size following the relation presented in equation 6.

Cohesion becomes significant for particles that are silt-sized and smaller due to surface charge effects, which increases the threshold for entrainment compared to predictions from the Shields curve (Kemper et al., 1987; Kothyari and Jain, 2008). As a result, sand is the most easily entrained material: larger particles are harder to move due to their mass, while smaller particles are harder to move due to cohesion. Of course, most stream banks are composed of mixtures of cohesive and non-cohesive sediments. The threshold entrainment stress for sand increases rapidly with increasing fraction of clay and silt, with reported increases of up to two orders of magnitude for clay-rich river banks (Kothyari and Jain, 2008). For gravel particles of order centimeter and larger, however, the entrainment stress varies little with the addition of clay and silt (Kothyari and Jain, 2008). Taken together, we naively expect that rivers with bed sediment $D_{50} > 10^{-2}m$ should organize to a threshold shear stress that is slightly in excess of the threshold predicted by the Shields curve. For natural rivers with bed sediment smaller than about a centimeter, cohesive sediments (if present) will lead to channel banks with entrainment thresholds that are larger than predicted by the Shields curve. The minimum threshold fluid stress for a sand-bed river is $\tau_b \sim 0.1 N/m^2$ based on the Shields curve. Without knowledge of bank materials in the data used here, we use results from a systematic study that examined the influence of silt-clay content on the erosion threshold of natural sandy river banks. Julian and Torres (2006) reported a maximum stress of $\tau_b \approx 25 N/m^2$ for banks composed entirely of silt and clay. For typical banks with silt-clay fractions of a few tens of percent, and/or moderate vegetation coverage, a typical value for the critical stress is $\tau_b \approx 5 N/m^2$ (Julian and Torres, 2006; Tal and Paola, 2007; Braudrick et al., 2009).

Turning to the global dataset, we compare the bankfull shear stress $\tau_{bf}$ to bed-sediment grain size $D_{50}$ for all rivers (Fig. 7). While there is significant scatter, we notice a general pattern in the data; sand-bed rivers show no relation between bankfull shear stress and bed-sediment grain size, while gravel-bed rivers exhibit increasing shear stress with grain size. Projecting the threshold stress based on the Shield curve onto the data, we see that gravel-bed rivers generally follow the curve while sandy rivers plot significantly above it. The range of $\tau_{bf}$ for sandy rivers overlaps with, and is slightly offset from, the range of threshold stresses reported for sand-mud mixtures (Fig. 7). The "typical value" of $\tau_b = 5 N/m^2$ runs through the middle of the sandy rivers. The threshold-limiting material may be assessed by comparing the threshold stress of mud-sand mixtures to the threshold stress determined from the Shields curve; we see that most rivers with $D_{50} > 1cm$ are limited by gravel mobility, while most rivers with $D_{50} < 1mm$ are limited by bank mobility (if cohesive sediment is present).

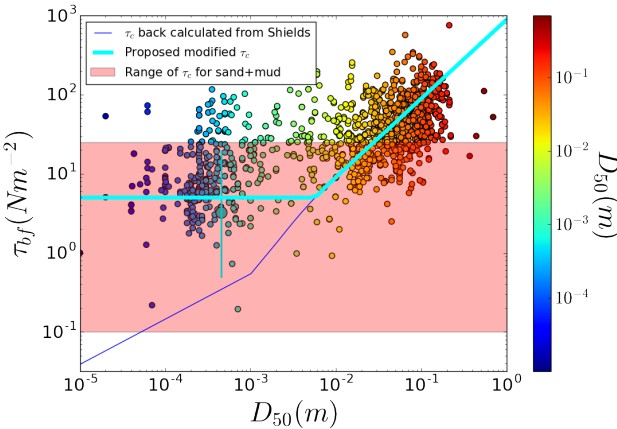

**Figure 7.** Potential adjustment of river-bed shear stress to the threshold-limiting material for the global data. The rising line from left to right indicates expected critical shear stress determined from grain size based on the Shields curve fit of van Rijn (2016). Gravel-bed rivers generally fall along this line, but sandy rivers generally plot significantly above it. Flat line shows a reference critical shear stress for the middle of the range of sand-mud mixtures. Cyan line indicates the trace of the threshold-limiting stress. For rivers with bed sediment grain sizes smaller than about a millimeter, we expect bank material to be threshold limiting; for gravel-bed rivers, the bed is expected to be threshold limiting.

The above trends provide tentative, albeit equivocal, support for the hypothesis that all alluvial rivers are near-threshold channels adjusted to the threshold-limiting material. For the case of gravel-bed rivers, this corresponds to a transport stage close to one for the bed material at bankfull. For sand-bed rivers with cohesive banks, we expect the transport stage of bed material to be roughly the ratio of the bank to bed entrainment thresholds, which could be in the range $10^0 \leq \tau_{*bf}/\tau_{*c} < 10^3$.

Because sand has the lowest threshold, and most natural river banks contain some cohesive materials, transport stage for sandy rivers is typically much greater than 1 leading to suspension channels. Given the paucity of alluvial river-beds with median grain sizes between 1 mm and 10 mm — the range where we expect cohesive banks to become important — these factors give rise to a bi-modal distribution of transport stage. In terms of hydraulic geometry, data indicate that cross-sectional area is controlled primarily by hydraulic conveyance as it has a very tight relation with bankfull discharge for all rivers. The partitioning of this

area into width and depth appears to be related to the threshold constraint imposed by bank-toe material.

We close this section with a brief but important aside on the distinction between hydraulic geometry and dynamics. The idea that all alluvial rivers are near threshold may at first seem incompatible with the intrinsic and incessant dynamics we observe: widening/narrowing, meandering, sorting, and bed/bar form evolution. In this context the (near-)threshold channel geometry is the statistically-expected behavior in a dynamic, stochastic system — analogous to a mean bed-load flux, or Reynolds averaging

in fluid mechanics — that does not represent system behavior at any particular instant (Furbish et al., 2016). The experimental findings of Reitz et al. (2014) make this point well: "Although individual channels in the braided river are constantly changing shape through scour and fill, these appear to be fluctuations around a robust [near-threshold] geometry that becomes apparent when many individual channel geometries are averaged together." Some of the scatter in hydraulic geometry scaling plots may

be due to a variety of factors such as: influences from vegetation, localized/temporary imbalances between the rate of floodplain formation and bank failure, and partial submergence of grains in the flow.

# 7 Conclusions

We propose that all alluvial rivers, regardless of their bed material grain size, organize their hydraulic geometry such that they cluster around the threshold of motion for the most resistant material — the structural component of the channel that is most difficult to mobilize. For coarse-grained rivers, the threshold-limiting material is the gravel that comprises the bed and bank toe. In contrast, the threshold-limiting material in sand-bedded rivers is not the bed material, but the cohesive mixture of mud and sand (and vegetation) that makes up the toe of the river bank. Thus, we posit that it is the difference in entrainment threshold between the non-cohesive bed and cohesive banks that facilitates suspended-sediment transport in sandy rivers. We expect that, in very fine-grained mud channels, the threshold-limiting material is the mud that makes up both the bed and the bank toe. Consideration of the slope- or grain-size-dependence of the critical Shields stress shows that alluvial rivers are bi-modal in terms of transport stage and bed-material grain size, and that these modes correspond generally (but not always) to bed-load gravel rivers and suspension sand rivers. We acknowledge, however, that other factors unaccounted for in our simple analysis must also play a role. For example, form drag due to roughness on multiple scales (grains, bed forms, bars, meanders) can drastically change the effective bed stress (Kean and Smith, 2006). We suspect that proper accounting of flow resistance would reveal a stronger signal of near-threshold organization. Of course, determination of the entrainment threshold at the bank toe is needed to provide direct confirmation of the hypothesis we propose here. Experiments have qualitatively demonstrated the influence of cohesion on channel geometry (Kothyari and Jain, 2008; Tal and Paola, 2007; Braudrick et al., 2009), but a systematic examination of channel shape as a function of increasing cohesion in sand-mud mixtures is necessary to demonstrate the viability of the threshold-limiting hypothesis.

*Author contributions.* K.B.J.D performed the research and analyzed the data, D.J.J. supervised the research, and both authors wrote the paper.

*Competing interests.* The authors declare no conflict of interest.

*Acknowledgements.* We thank Michael Singer and Olivier Devauchelle for sharing their data for the Sacramento River and seepage erosion channels in Florida, respectively, and James Pizzuto for useful comments that helped to frame aspects of this paper. Research was sponsored by the Army Research Laboratory and was accomplished under Grant Number W911-NF-16-1-0290. The views and conclusions contained in this document are those of the authors and should not be interpreted as representing the official policies, either expressed or implied, of the Army Research Laboratory or the U.S. Government. The U.S. Government is authorized to reproduce and distribute reprints for Government purposes notwithstanding any copyright notation herein.

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
