# Peer review of "Evidence of, and a Proposed Explanation for, Bimodal Transport States in Alluvial Rivers"

_Earth Surface Dynamics, 2017_

## Referee Comment (RC1) · A. Wickert (Referee) · 10 Dec 2017

Dunne and Jerolmack have provided a concicse and timely review of the literature on the threshold conditions that control river width and sediment transport. They then extend this work into a hypothesis for how sand-bed rivers behave – and I believe their approach to be solidly on the right track to solving this problem. Their analysis of the now-available data sets shows the clear bimodal distribution in river morphology and sediment transport – whether it is a gravel-bed channel at the threshold of motion or a sand-bed channel with cohesive banks. This extension of the classic Parker (1978) analysis to the question of particle-mass-controlled banks to cohesion-controlled banks

is central to understanding not only the form of alluvial and (more generally) transport-limited rivers, but also their sediment transport rates, dynamics, and time-scales and mechanisms of response to perturbation.

This is the best manuscript that I have reviewed this year, and I reccomend it for publication. The following comments are mostly editorial; a couple ask for further explanation, but the paper remains sound without their being answered.

General:

Use `\citep` instead of `\citet` with extra parentheses

Line-by-line:

p.1,l.17-19: I find Tal and Paola (2007) to be a key reference for meandering vs. braiding when involving vegetaion; perhaps it is just not cited due to the many others here, but I suggest that you read it if you haven't!

Equation 1: Consider splitting onto 3 rows for better readability

p.2,l.1: remove indent

Fig. 1: Why does this schematic have to have different bed and bank material? In particular, your statement that the center of the channel is only slightly in excess of critical Shields stress indicates that it is more likely all gravel that you are trying to illustrate. Do you intend to use this for both the gravel-bed and sand-bed channel examples? If so, just a little bit of clarification/generalization will be needed. (As a graphic, this is an excellent block illustration/generalization of the Parker (1978) line drawing, by the way, and one that I'd like to borrow for lectures.)

p.3,l.1: Parker (1978a) predicted... (no need for "Parker's model")

p.3, l.22-24. Note also that Pfeiffer et al. (2017) demonstrate that gravel-bed rivers are significantly above the threshold of motion in rapidly-uplifting settings. They argue that this is due to armoring, though I have my own ideas about this (as-of-yet unpublished,

and therefore nothing that you'll have to tangle with in a review).

p.3,l.24: remove comma after "slope"

Fig. 2: Show legend in only one panel to not overlap with data

Fig. 3: It looks in the figure like you have one point from seepage channels and one fromm experiments. Therefore, I suggest that you reword "River channels ..." to take into account that this is not quite so plural, and is not tested to be generalizable.

p.8,l5-6: Which site-specific variations are you considering, and how/why are they important?

Fig. 5: This is to me, thus far, the figure of the paper.

p.10,l.19-25: Yes, I completely agree, and have spent some time thinking about this myself.

p.11,l.1-3: Could you discuss other reasons for cohesion? Interlocking grains and capillary forces come to mind. Are these significant compared to the surface charge effects?

p.11,l.3: material; –> material:

p.11,l.31-32: Would you like to discuss some of the reasons for the low frequency of channels with 1-10 mm grain size? In particular, do you think that this may have to do with the crystal size / granule break-down problem, or possibly be connected to the transition between cohesion-dominated banks and particle-weight dominated banks that makes these grains either difficult to move or whisked away in a larger-clast gravel-bed river? This is of course ignoring arguments for equal mobility...

Figure 7: I have a sketch that looks quite a lot like this in my notes: I also think that this is how the system must work, in general, with some of the details hidden in the scatter.

---

## Referee Comment (RC2) · M.G. Kleinhans (Referee) · 23 Dec 2017

The manuscript addresses a question that emerged in the past half century from studies of hydraulic geometry sensu largo (Parker et al. 2007, 2008) and downstream fining (see Frings 2008 for review). The two main claims are: 1) that gravel-bed river have sediment mobility near the threshold for motion and sand-bed rivers are in the suspension regime; 2) that channel geometry is a result of the threshold of bed material in gravel-bed rivers and of cohesive bank material for sand-bed rivers. These claims are based on reanalysis of existing hydraulic geometry data and data of rivers along the gravel-sand transition.

[Figure]

The manuscript is mostly based on review and replotting of existing data. The manuscript is clearly written and the data is well-presented in the figures. However, the manuscript is flawed in a number of aspects: a large number of essential references are missing in the presented datasets and in the text; the insights presented are not novel and that based on a downstream fining river is flawed; some of the reasoning about experiments is based on only a few points while many more are available in the literature. Below these points are detailed.

It is not exactly clear what precisely the argument of the authors is. It seems to be that "there are mainly sand-bed, suspension-dominated rivers and gravel-bed, bedload-dominated rivers because of the scarcity of the intermediate sizes on this planet and because the typically available floodplain materials are strong enough to withstand the typical bed shear stress in sand-bed rivers but not in gravel-bed rivers." The arguments revolves around the scarcity of sediments between sand and gravel, discussed by the authors, and the scarcity of sediments between sand and floodplain fines, not discussed by the authors. If this is correctly represented then these ideas have already been published in many papers over the past decades and are therefore not novel. If not correct then the argument is not clearly enough presented.

An elegant way, in principle, to look at the relation between bed mobility and channel geometry is the dataset of Singer (2010). This shows a gradual downstream reduction in bed shear stress but a zone with a bimodal reduction in median particle size and in excess mobility due to patchiness of bed material (also found in Paola and Seal 1995). However, the authors here invoke this observation to argue for bimodal behaviour in the entire dataset and this argument is flawed. Here, and on P3 L14-16, important references are missing, such as Yatsu (1955), Paola and Seal (1995) and work by Frings (review in 2008). The large body of literature on downstream fining bracketed by these references show three things. First, many fluvial sediments are bimodal because of the nature of sediment and the manner in which it wears down. The effect is that there are very few rivers with dominantly fine gravel in their bed. Second, in a bimodal sediment distribution where the two modes gradually change in height, as in downstream fining where the finer mode increases in abundance as the coarser mode reduces, the median grain size flops suddenly from one mode to the other where the mean size would not. This means that the representation of downstream fining by the median size is misleading the authors to believe in the bimodal behaviour. Third, many gravel-sand transitions are rather gradual, except in many relatively small rivers (Frings 2011). This is partly due to patchiness, bend sorting, and the transition from clast-supported to matrix-supported sediments. Elsewhere it has also been argued that many rivers show a third mode of sediment: the silt and clay that form the floodplains. Usually the material between that mode and the sand mode is scarce too (see Kleinhans 2010 for review).

The systematic trend of increasing bankfull Shields number with decreasing grain size is disputed on the basis of two pieces of evidence related to published experiments. The first is that seepage channels formed in experiments with sand cluster at the threshold for motion, and the second is that other sand-bed channels in the laboratory likewise are at the threshold for motion. But this misses the point of scaling in experiments entirely. The Shields number is the relevant scale for sediment mobility, and in experiments that scale down the size of systems in nature by orders of magnitude, of course the sand is near the threshold for motion because it simulates gravel and even at sand size needs steeper slopes than in reality. This is why other experiments with self-formed seepage channels were conducted with low-density sediment (Marra et al. 2014), and consequently the mobility ranged from 0.044-0.68.

Furthermore, there are a large number of other experiments with self-formed channels (compiled in Kleinhans et al. 2014 Fig. 9 and Table 1 and some more in 2015) that show a range of mobilities in the context of a large river dataset that also show a gradual transition from bedload to suspended load dominated behaviour. This includes high-mobility experiments with sand and floodplain-forming low-density sediment (Paola group, Minneapolis) and with materials that strengthen banks (famous experiments of Hoyal, ExxonMobil) (see Kleinhans et al. 2015 for references). The main point is that the nature of the bank material determines the channel geometry. This is essential for bar and river patterns and was incorporated in the hydraulic geometry in Kleinhans et al. (2015) because correct scaling of the bank material in experiments leads to desired channel aspect ratios. The qualitative understanding behind this work goes back at least to Ferguson (1987), and shows that the experimentally inferred relation of the straight-meandering-gravel transition as a function of slope by Schumm and Kahn (1972) is wrong and the bank stability is key. Furthermore, the natural and experimental channels adhere well enough to the hydraulic geometry relations of Parker et al. (2007) when accounted for the strength of the banks (Kleinhans et al. 2015), which allowed application of the Parker relation to bar theory. It is already well-known that the absolute dimensional shear stress in gravel-bed rivers is much larger than that in sand-bed rivers despite the lower sediment mobility, so that bank material unstable in the gravel reach can be stable in the sand-bed reach. The hypothesis on P10 L14-16 has already been voiced in a large body of bank erosion literature underlying the BSTEM model by Simon et al. (2000 and later references). This argument has not become more precise by the scatter plot in Fig. 7. Also the authors do not present new data or new insights; for example P10 L27-32 states that information about bank material is usually lacking and this is precisely why Kleinhans and van den Berg (2011) had to introduce that counterintuitive streampower measure and why Eaton and Church (2007) developed a rational regime theory that can be used to invert to bank stability measures from observed channel dimensions. These papers and many others already make that point; however, these references are lacking here.

The final remark of the discussion is somewhat simplistic: "Some of the scatter in hydraulic geometry scaling plots may be due to stochastic fluctuations around the mean behavior." Here, stochastic is a rather dirty term for a number of well-known sources of variability. Apart from the fact that vegetation creates its own patterns with its own scales on the banks, leading to variations in channel dimensions, the balance between bank erosion and floodplain formation is not always in equilibrium because of

the specific length and time scales of bank failure and floodplain / levee formation. The assumption of the authors is that measured channel dimensions in their compiled dataset are due to such variations, but it is quite likely that the original authors of the datasets underlying this compilation already averaged out such variation. In fact, this is certainly the case for the Kleinhans and van den Berg (2011) dataset and in the experimental datasets from our laboratory, that were also taken up in the compilation of Metivier, and are therefore possibly part of the present compilation.

Finally, the authors present their data online, which is commendable, but entirely unreferenced, which is unacceptable, not only because it does not pay proper credit to the original owners. These data originate from many sources and likely the workers used somewhat different methods which may strongly affect the scatter and trends. Furthermore, the seepage and experimental datasets are sorely lacking numbers, which are available in the recent literature.

References

Eaton, B. C., and M. Church (2007), Predicting downstream hydraulic geometry: A test of rational regime theory, J. Geophys. Res., 112, F03025, doi:10.1029/2006JF000734.

Ferguson, R., 1987. Hydraulic and sedimentary controls of channel pattern. In: Richards, K. (Ed.), River Channels: Environment and Process. Blackwell, Oxford, UK, pp. 129-158.

Frings, R.M. (2008), Downstream fining in large sand-bed rivers, Earth-Science Reviews 87, 39-60, doi:10.1016/j.earscirev.2007.10.001.

Frings, R.M. (2011), Sedimentary characteristics of the gravel-sand transition in the River Rhine, J. of Sedimentary Research 81, 52-63.

Kleinhans, M.G. (2010), Sorting out river channel patterns, Progress in Physical Geography 34(3), 287-326, doi:10.1177/0309133310365300

Kleinhans, M.G. and J.H. van den Berg (2011), River channel and bar patterns explained and predicted by an empirical and a physics-based method, Earth Surf. Process. Landforms 36, 721-738, doi:10.1002/esp.2090.

Kleinhans, M.G., W.M. van Dijk, W.I. van de Lageweg, D.C.J.D. Hoyal, H. Markies, M. van Maarseveen, C. Roosendaal, W. van Weesep, D. van Breemen, R. Hoendervoogt and N. Cheshier (2014), Quantifiable effectiveness of experimental scaling of river- and delta morphodynamics and stratigraphy, Earth-Science Reviews 133, 43-61, doi:10.1016/j.earscirev.2014.03.001

Kleinhans, M.G., C. Braudrick, W.M. van Dijk, W.I. van de Lageweg, R. Teske and M. van Oorschot (2015). Swiftness of biomorphodynamics in Lilliput- to Giant-sized rivers and deltas, Geomorphology 244, 56-73, doi:10.1016/j.geomorph.2015.04.022

Marra, W.A., L. Braat, A.W. Baar and M.G. Kleinhans (2014). Valley formation by groundwater seepage, pressurized groundwater outbursts and crater-lake overflow in flume experiments with implications for Mars. Icarus 232, 97-117, doi:10.1016/j.icarus.2013.12.026.

Marra, W.A., S.J. McLelland, D.R. Parsons, B.J. Murphy, E. Hauber and M.G. Kleinhans (2015). Groundwater seepage landscapes from distant and local sources in experiments and on Mars. Earth Surface Dynamics 3, 389-408, doi:10.5194/esurf-3-389-2015

Paola, C. and R. Seal (1995), Grain size patchiness as a cause of selective deposition and downstream fining, Water Resources Research 31(5), 1395-1407.

Parker, G., P. R. Wilcock, C. Paola, W. E. Dietrich, and J. Pitlick (2007), Physical basis for quasi-universal relations describing bankfull hydraulic geometry of single-thread gravel bed rivers, J. Geophys. Res., 112, F04005, doi:10.1029/2006JF000549.

Parker, G., T. Muto, Y. Akamatsu, W.E. Dietrich and J.W. Lauer (2008), Unravelling the conundrum of river response to rising sea-level from laboratory to field. Part II. The Fly-Strickland River system, Papua New Guinea, Sedimentology 55, 1657-1686, doi:

10.1111/j.1365-3091.2008.00962.x.

Schumm, S., Khan, H. (1972), Experimental study of channel patterns. Geol. Soc. Am. Bull. 83, 1755-1770.

Simon, A., Curini, A., Darby, S., Langendoen, E., 2000. Bank and near-bank processes in an incised channel. Geomorphology 35, 183-217.

Yatsu, E., (1955), On the longitudinal profile of the graded river: American Geophysical Union, Transactions, v. 36, 655-663.

**ESurfD**

---

## Referee Comment (RC3) · Anonymous Referee #3 · 7 Jan 2018

I have reviewed the paper, and the subject is quite an interesting one. However, the analysis seems to be fundamentally flawed to me, and ignores a vast body of work that has been trying to make sense of reach-scale channel geometry for many decades. The paper relies far too heavily on work done by a small group of authors. As a result, much of the insight that the paper does produce I feel has already been better explored and presented by previous authors. Because this paper is not based on a fair and balanced understanding of the previous work, it is not suitable for publication, in my opinion.

Critical Issues: On page 5, line 10, you state that you assume the Chezy friction factor

is constant. I can see no possible justification for this assumption. Looking at Ferguson 2007 and Eaton and Church 2011, it is clear that Cf varies dramatically with relative roughness, and that changes in relative roughness also produce order of magnitude changes in the bedload sediment concentration generated at the same critical dimensionless shear stress. These scale variations are fundamental, and need to be included in the numerical analysis. This is one of the reasons that Millar took the numerical approximation approach that he did, and then generated power law equations using a Monte Carlo modelling approach (see Millar, 2005). This assumption appears to me to fundamentally undercut the entire following analysis.

Specific comments: Page 1, Line 15: you claim that the empirical hydraulic geometry equation are robust, and show remarkably little variation in the exponents: this is a stretch. log log plots hide the true scale of differences between datasets; the lumped data includes systems where W/d ratio declines downstream, and where it increases downstream; it includes systems where depth is nearly constant, and ones where it changes as expected. You also fail to observe that simple Froude scaling nearly explains the trends that are found in nature, which indicates that most of the variance can be explained simply by the size of the system, not any particular organizing principle. This is well described by Eaton in the Hydraulic Geometry chapter of the Treatise on Geomorphology. In fact, there are significant variations in the exponents, when they are compared against the Froude scaling exponents, and I believe this is well described by Eaton and Church (2007) in JGR.

Page 1, Line 20: you attribute the concept of the dimensionless discharge to Metivier (2016). This is certainly not the primary source for a dimensionless discharge. Andrews used it in his 1984 paper (though there is an unfortunate typo in the final version, he used exactly this definition of a dimensionless discharge).

Page 2, Line 1 It is remarkable that you do not refer to Ferguson's classic work on the basis for regime theory. I believe that this is still one of the best summaries of the problem, and provides a much more satisfactory statement of the problem than the

authors provide here.

Page 2, line 10: The "ground state" approach by Metivier sound exactly like the threshold channel approach used by many previous authors, including Lane, 1955; Henderson, 1966; Stevens, 1989.

Page 2, line 2: the discussion of the stable channel paradox seems to be to miss the point. The key thing to realize is that the shear stress acting on channel banks is lower than the average boundary shear stress, and that the shear stress acting on the bed is higher than average. This particular issue was very well addressed by Rob Millar's implementation of a regime theory that considered bank strength, using the shear stress partitioning approach by Knight, Flintham and Carling. I recommend reading Millar 2005 and the numerous relevant papers cited therein. I believe that Millar's contribution has adequately asked and answered the questions that this paper attempts to answer, and that the data analysis presented herein does not provide any additional insight to the problem. In any case, I do not see the justification for publishing this analysis without acknowledging the previous work!

page 3, line 11: you state that there is no accepted model for the equilibrium geometry of rivers far above threshold. I think is is an unfair and inaccurate representation of the current state of the science. There are models (like Millar and Quick, 1998) that successfully predict the geometry of such streams. They are published, and have been successfully tested, yet you appear to disregard all of these models without even bothering to mention them.

Analysis in Fig 2: does this really tell us anything that we do not already know from Church's (2006) exploration of the plotting positions of various rivers? While the plots are somewhat different, I do not see what novel insight they provide, particularly given the scale distortions introduced by the inappropriate assumption that $C_f$ is constant!

Page 6, lines 1 to 5: You text is not an accurate representation of the bank strength issue. With respect to vegetation, the effect on relative bank strength is fundamentally

scale dependent (see Eaton and Giles, 2009, Eaton and Millar, 2017), which you fail to mention (and which also introduces significant scale distortions), and the effect on channel with is close to a linear one. As banks become very erosion resistant, then changes in width are reduced, because the channel is able to reach the hydraulically optimal form, beyond with narrowing does not produce any increase in bed shear stress (basically this is the geometry predicted by Wobus 2004 for erosion into bedrock).

Page 6, line 9: your use of Schumm's M findings is a poor choice, since that analysis is a classic tautology. Clay and silt are never found on the channel bed, so Schumm's index (which is the percent silt and clay averaged over the entire channel boundary) builds in the width depth ratio; therefore it cannot be used to make a meaningful prediction of the width depth ratio. Simons and Albertson (1963) do manage to make some progress on the sedimentological controls for canals, however.

Page 6, line 15: the authors present a hypothesis about what controls bank erosion, but that hypothesis was advanced previously by Nanson and Hickin, and is the basis for the modesl developed by Millar and Quick (1993) and by Eaton (2006).

---

## Editor Comment (EC1) · JM Turowski (Editor) · 8 Jan 2018

Dear authors,

we have now received three reviews for your manuscript. One is very supportive and has only minor (mainly editorial queries). The other two reviews are closely aligned in their criticism. There are two main points.

First, both reviewers point out that a large body of relevant literature has been overlooked, which should be acknowledged and discussed. I agree with this point. The reviewers have pointed out a number of relevant papers. I add to these the recent work

of Blom et al. (two GRL papers in 2017 and one JGR paper in 2017), who discuss gravel-sand transitions, equilibrium and quasi-equilibrium states and the effects of discharge variability, and of Pfeiffer et al. (PNAS 2017), who investigated the morphology of gravel bed channels in the USA. In addition, the authors may want to look at the literature on steep streams – there has been recent work on the interplay of bed roughness, flow hydraulics and sediment transport that may be informative for some of the discussion. A comprehensive study on flow velocity has been published for example by Rickenmann and Recking (WRR 2011), and on bedload transport for example by Schneider et al. (WRR 2015). There may be other relevant literature that has not been mentioned in the reviews or by me – the newly provided references may be a good starting point for a wider research.

Second, both reviewers fail to see the novelty in the work. I ask you to clearly delimit where the paper is a mere review of published results and where you go beyond what has been done before, in particular in light of the additional literature mentioned above.

There are a large number of other critical points by the reviewers, which I ask you to address in detail. In particular, I would like to highlight the closing comment of reviewer #2. You have worked with data measured and compiled by many other scientists, and these should be duly acknowledged. The least that should be done is the addition of stream and site names, coordinates, and suitable references in the data tables provided in the supplementary material. Otherwise, researchers will have little chance to scrutinize your data and your results in the future.

With best wishes, Jens Turowski

---

## Editor Comment (EC2) · JM Turowski (Editor) · 15 Feb 2018

Dear Kieran,

the figures in your reply do not seem to be correctly displayed, at least not in the .pdf that I can access. Could you upload a corrected version?

Thanks, Jens
* * *

---

## Author Comment (AC1) · 15 Feb 2018

p.1,l.17-19: I find Tal and Paola (2007) to be a key reference for meandering vs. braiding when involving vegetaion; perhaps it is just not cited due to the many others here, but I suggest that you read it if you haven't!

We do cite Tal and Paola 2007.

Equation 1: Consider splitting onto 3 rows for better readability

Done

ESurfD
p.2,l.1: remove indent

Done

Fig. 1: Why does this schematic have to have different bed and bank material? In particular, your statement that the center of the channel is only slightly in excess of critical Shields stress indicates that it is more likely all gravel that you are trying to illustrate. Do you intend to use this for both the gravel-bed and sand-bed channel examples? If so, just a little bit of clarification/generalization will be needed. (As a graphic, this is an excellent block illustration/generalization of the Parker (1978) line drawing, by the way, and one that I'd like to borrow for lectures.)

This is meant to illustrate a sand-bedded river. we have added clarification.

p.3,l.1: Parker (1978a) predicted... (no need for "Parker's model")

Done

p.3, l.22-24. Note also that Pfeiffer et al. (2017) demonstrate that gravel-bed rivers are significantly above the threshold of motion in rapidly-uplifting settings. They argue that this is due to armoring, though I have my own ideas about this (as-of-yet unpublished, and therefore nothing that you'll have to tangle with in a review). We, and others, have attempted to replicate the results of the Pfeiffer et al. paper with no success. Their
results of their work are dependent upon the biases implicit in the assumption of slope-dependent critical shields stress. As their results cannot be replicated under more rigorous testing, we will not use that paper as a reference. See comments above to the Editor on this.

p.3,l.24: remove comma after "slope"

Done

Fig. 2: Show legend in only one panel to not overlap with data

Done

Fig. 3: It looks in the figure like you have one point from seepage channels and one from experiments. Therefore, I suggest that you reword "River channels ..." to take into account that this is not quite so plural, and is not tested to be generalizable.

The single, larger points are meant to illustrate the mean of multiple measurements taken along a single longitudinal profile or multiple runs in a laboratory setup.

p.8,l5-6: Which site-specific variations are you considering, and how/why are they important?

The term "site-specific" was meant to refer to the grain size and slope. We merely

wanted to state that there are multiple ways of estimating critical.

p.8,l5-6: Which site-specific variations are you considering, and how/why are they important?

The term "site-specific" was meant to refer to the grain size and slope. We merely wanted to state that there are multiple ways of estimating critical.

p.11,l.1-3: Could you discuss other reasons for cohesion? Interlocking grains and capillary forces come to mind. Are these significant compared to the surface charge effects

Capillary forces would not be a factor because the bank toe is perennially saturated. We hypothesize that part of the reason for the huge range in bank critical shear stresses from previous work is because previous studies sampled from a variety of locations up the bank, which would then incorporate capillary forces despite those not having an effect at the bank toe. We do not believe that the interlocking of grains is a significant cohesive factor because grain interlocking would happen in both cohesive and non-cohesive settings, however we see that for non-cohesive systems rivers adjust themselves to the threshold of motion. Perhaps grain interlocking is implicitly accounted for in this clustering around the threshold of motion.

p.11,l.3: material; –> material:

Done

p.11,l.31-32: Would you like to discuss some of the reasons for the low frequency of channels with 1-10 mm grain size? In particular, do you think that this may have to do with the crystal size / granule break-down problem, or possibly be connected to the transition between cohesion-dominated banks and particle-weight dominated banks that makes these grains either difficult to move or whisked away in a larger-clast gravel-bed river? This is of course ignoring arguments for equal mobility...

While this is indeed an interesting question, we do not believe the grain size gap found in channels is of particular relevance to this paper.

---

## Author Comment (AC2) · 15 Feb 2018

[esurf, manuscript]copernicus [utf8]inputenc color

**1   Jens Turowski**

First, both reviewers point out that a large body of relevant literature has been over-looked, which should be acknowledged and discussed. I agree with this point. The reviewers have pointed out a number of relevant papers. I add to these the recent work

of Blom et al. (two GRL papers in 2017 and one JGR paper in 2017), who discuss gravel-sand transitions, equilibrium and quasi-equilibrium states and the effects of discharge variability, and of Pfeiffer et al. (PNAS 2017), who investigated the morphology of gravel bed channels in the USA. In addition, the authors may want to look at the literature on steep streams – there has been recent work on the interplay of bed roughness, flow hydraulics and sediment transport that may be informative for some of the discussion. A comprehensive study on flow velocity has been published for example by Rickenmann and Recking (WRR 2011), and on bedload transport for example by Schneider et al. (WRR 2015). There may be other relevant literature that has not been mentioned in the reviews or by me – the newly provided references may be a good starting point for a wider research.

We thank the editor for his comments and suggestions. We have a new and expanded introduction that encapsulates previous work/thinking on equilibrium channel geometry — especially on rational regime theory, influence of cohesion on channel width, and related approaches. We have a few points of contention related to the references suggested by the editor:

1. We are aware and appreciative of the work done by Blom et al. Indeed, several theoretical and empirical constraints used by Blom and colleagues to develop their recent analytical models utilize work from our group (Jerolmack and Brzinski, Geology 2010; Miller, Retiz and Jerolmack, GRL 2014; Phillips and Jerolmack, Science, 2016). It is not the intent of this paper, however, to examine discharge variability or other effects. I note that Blom and colleagues' treatment of the gravel-sand transition is similar to our treatment here, so there is no discrepancy. This work is now cited, but a deeper comparison is not really warranted in our opinion.

2. We, and others, have attempted to replicate the results of the Pfeiffer et al. paper with no success. First of all, their statistical treatment of the data is insufficient

to demonstrate a real effect of sediment supply *at best*; at worst, their statistical methods are faulty. Moreover, their principle conclusion is dependent upon the biases implicit in the assumption of slope-dependent critical shields stress; for rivers in their data set where critical Shields stress was actually independently constrained, these values indicate rivers only slightly above threshold. Finally, those authors did not actually measure or report hydraulic geometry or discharge (despite the paper title) so one cannot assess whether the hydraulic geometry scaling of their channels is distinct from others. We do not wish to debate the merits of that paper this one. In addition, the main point here is that rivers are *bimodal* in their transport stage, because gravel-bed rivers don't care about bank material but sand-bed rivers do. Even if the *slightly* higher values for critical Shields stress in the Pfeiffer et al. paper were real, these data would still cluster with the near-threshold gravel. None of that influences our conclusions.

3. While we do agree that the inclusion of steep stream data in the global data set being analyzed would potentially provide interesting insights into the variance within the coarse-grained portion of the data set, the point of this paper is not to provide illumination into this aspect of the science. The objective of this paper, which, upon suggestion from the editor, has been more clearly delineated in the paper, is to test whether or not the bimodality of sediment transport states survives in light of evidence to the contrary and to provide a hypothesis as to why or why not. We do not believe that the inclusion of steep stream data and further analysis of flow velocity or bed roughness will assist in this endeavor. Note that we have not explicitly excluded steep streams; many in our dataset are up to and even above ten percent. It is likely that flow resistance and other factors correspond to the variance of transport stages seen within gravel-bed river data. But the main point of this paper not the variance within either near- or far-above threshold regimes; rather, it is the separation of data into these two regimes, and its relation to when critical stress for bed material drops below that of the banks.

4. Overall, we want to properly cite the previous literature that is relevant to our study, but we don't think it is useful to exhaustively review all work done on equilibrium channel geometry. This paper is about bi-modal states, and a potential explanation for the two modes; it is not about what happens within each mode.

Second, both reviewers fail to see the novelty in the work. I ask you to clearly delimit where the paper is a mere review of published results and where you go beyond what has been done before, in particular in light of the additional literature mentioned above

It is evident that we have failed to clearly state where our work fits into the wealth of pre-existing work on regime theory. Thus, we add the following clarification to our manuscript:

"Regime theory" is the application of these agreed upon relationships with the addition of one additional threshold channel based-assumption to allow for closure. There are three dominant branches of regime theory, each with their own form of a threshold channel closure assumption that separate regime theory into three distinct schools of thought: 1) assume that river are canals, and thus threshold channels; 2) assume that the transport regime is purely bedload and solve the 2-D flow field to balance fluid shear stress and particle weight at the edge of the channel, while simultaneously allowing for transport at the center; 3) assume that the river undergoes an optimization process that maximizes friction in order to reduce fluid shear stress, ultimately resulting in a threshold channel.

The first school of thought is based upon work done to calculate the shape of a stable canal for which the bed material is at the threshold of motion (Glover and Florey (1951)). This work has been extended to natural rivers by Henderson (1961), and offers an explanation for observations of alluvial river width relating to the water discharge

(Henderson (1961); Andrews (1984); Metivier et al. (2007)). This line of thinking links well in with the second branch of a regime theory which as was established by Parker (1978a) which solved the 2-D stress field to show that, for a pure bedload river, the channel is at the threshold of motion for the material at the banks and slightly above the threshold of motion in the center, allowing for the river to transport sediment, while at the same time maintaining a stable and consistent width. This model is supported by both global compilations of data and case studies of individual rivers that demonstrate that gravel-bedded rivers that translate their sediment load as bedload are slightly off-set from a threshold of channel (Philipps and Jerolmack (2016); Gaurav et al. (2015); Metivier et al. (2016)). Parallel to this grain size-dependent channel geometry is the concept of optimization which assumes that rivers seek a threshold channel condi-tion by maximizing the flow resistance within the channel to minimize the fluid shear stress (Eaton and Church (2004); Eaton and Churh (2007)). The rational regime the-ory put forward by Eaton attempts to infer the importance of bank strength given devia-tions away from the threshold condition that is posited by optimality theory (Eaton and Church (2004); Eaton and Church (2007)), however they are predominantly calibrated on coarse-grained rivers where research has shown that the influence of cohesive mud is a minor control on the erodibility compared to the weight of the gravel (Kothyari and Jain (2008)). What distinguishes our work from this work is that we extend the concept of Parker's threshold channel model into the space occupied by fine-grained rivers by the suggestion that river channel geometries, and their subsequent sediment transport state are either controlled by the erodibility of their beds or their banks. This paper shows the transition from rivers that can be explained entirely by Parker's theory (i.e. channel beds and banks composed of uniform material transported entirely in bedload) to channels that cannot. For natural rivers, this transition most frequently occurs at the transition from a gravel-bedded to a sand-bedded condition. This transition coincides with the point at which bed material becomes small enough such that the cohesion of channel banks should become important. What we show is the that sediment trans-port state is bimodal because grain size is bimodal; the coarser gravel mode is more

difficult to entrain than any cohesive bank material, while the finer sand mode is easier to entrain than any cohesive bank material (if present).

[NOTE TO THE EDITOR: NO ONE HAS PROPOSED THIS, THAT WE ARE AWARE OF; THIS IS THE POINT OF OUR PAPER - NOTHING MORE, NOTHING LESS.]

There are a large number of other critical points by the reviewers, which I ask you to address in detail. In particular, I would like to highlight the closing comment of reviewer 2. You have worked with data measured and compiled by many other scientists, and these should be duly acknowledged. The least that should be done is the addition of stream and site names, coordinates, and suitable references in the data tables provided in the supplementary material. Otherwise, researchers will have little chance to scrutinize your data and your results in the future.

We thank the editor and reviewer 2 for this suggestion and have done so. The objective of the original formatting was the provide a "trimmed down" version of the cited data sets to provide only the data that was of immediate relevance to the analysis done in this paper. We have re-included the suggested information.

**2   Maartin Kleinhans**

It is not exactly clear what precisely the argument of the authors is. It seems to be that "there are mainly sand-bed, suspension-dominated rivers and gravel-bed, bed-load dominated rivers because of the scarcity of the intermediate sizes on this planet and because the typically available floodplain materials are strong enough to withstand the typical bed shear stress in sand-bed rivers but not in gravel-bed rivers." The arguments revolves around the scarcity of sediments between sand and gravel, discussed by the authors, and the scarcity of sediments between sand and floodplain fines, not

discussed by the authors. If this is correctly represented then these ideas have already been published in many papers over the past decades and are therefore not novel. If not correct then the argument is not clearly enough presented.

We hope that the newly included paragraph on regime theory addresses what our argument is. We agree with the reviewer that the statement he makes has been done before. We disagree that this is the main conclusion of our paper. We agree (of course, since one of us has worked a lot on this problem!) that river-bed grain sizes are bimodal; that forms one important piece of this puzzle. Bi-modal sediments are necessary *but not sufficient* to produce bi-modal transport states. What we show here is there is a second constraint; cohesive bank materials. When channel beds are composed of gravel they don't care about the banks, which are always easier to entrain; here we revert to a Parker-like condition of the bed-load river (and any variants of it). Only when beds are composed of sand do the banks matter, because sand is the easiest to entrain material; it is only when sand beds are combined with cohesive banks that rivers depart from Parker-type bedload river predictions. We are not aware of any paper that advocated for this position.

[Figure]

An elegant way, in principle, to look at the relation between bed mobility and channel geometry is the dataset of Singer (2010). This shows a gradual downstream reduction in bed shear stress but a zone with a bimodal reduction in median particle size and in excess mobility due to patchiness of bed material (also found in Paola and Seal 1995). However, the authors here invoke this observation to argue for bimodal behaviour in the entire dataset and this argument is flawed. Here, and on P3 L14-16, important references are missing, such as Yatsu (1955), Paola and Seal (1995) and work by Frings (review in 2008). The large body of literature on downstream fining bracketed by these references show three things. First, many fluvial sediments are bimodal because of the nature of sediment and the manner in which it wears down. The effect is that there

are very few rivers with dominantly fine gravel in their bed. Second, in a bimodal sediment distribution where the two modes gradually change in height, as in downstream fining where the finer mode increases in abundance as the coarser mode reduces, the median grain size flops suddenly from one mode to the other where the mean size would not. This means that the representation of downstream fining by the median size is misleading the authors to believe in the bimodal behaviour. Third, many gravel-sand transitions are rather gradual, except in many relatively small rivers (Frings 2011). This is partly due to patchiness, bend sorting, and the transition from clast-supported to matrix-supported sediments. Elsewhere it has also been argued that many rivers show a third mode of sediment: the silt and clay that form the floodplains. Usually the material between that mode and the sand mode is scarce too (see Kleinhans 2010 for review)

It is unclear to us what the problem is here. There is no point of disagreement between the reviewer's statement above and our position. First, the idea that "important references are missing". We can always add more references, and so we have. However, that misses the point; we are not trying to conduct an exhaustive review of the gravel-sand transition; a topic that one of us is well aware of having worked on it for quite some time. Yes, we are aware that sediment is often bimodal because of how it wears down. Indeed, we are presuming bimodality - and we demonstrate it with the data not as a "new" topic but just to verify that point. As for the second point of the reviewer, it is unclear what he is advocating for. He refers us to Paola and Seal — which is of course all about patchiness in the gravel-sand regime, basically showing that gravel and sand segregate into patches. But then the reviewer makes the point that the sudden switching of grain size is an artifact of choosing the median value — and by extension that river sediments are not patchy. Besides, this issue does not contradict the premise of the paper; whether the transition itself is patchy, flickering and rapid or more gradual and smooth, the main result is that within a river we see the same behavior that we see with the whole data set. *That* is the point of the Singer data; not to rehash the entire gravel-sand transition problem, just to show that one river does the same thing as the

global dataset, showing that patterns in the global data are not merely some artifact of data treatment. Finally, for the third point; it doesn't really matter how gradual or abrupt the gravel-sand transition is. That said, the reviewer fails to point out a paper that one of us co-authored (Miller, Reitz and Jerolmack) where we showed a collapse of sorting profiles across the gravel-sand transition for rivers big and small. The upshot is that the length of the gravel-sand transition is roughly ten percent of the length of the upstream gravel reach. Interesting? Maybe. Universal? Maybe not. Important for our analysis here? No.

The systematic trend of increasing bankfull Shields number with decreasing grain size is disputed on the basis of two pieces of evidence related to published experiments. The first is that seepage channels formed in experiments with sand cluster at the threshold for motion, and the second is that other sand-bed channels in the laboratory likewise are at the threshold for motion. But this misses the point of scaling in experiments entirely. The Shields number is the relevant scale for sediment mobility, and in experiments that scale down the size of systems in nature by orders of magnitude, of course the sand is near the threshold for motion because it simulates gravel and even at sand size needs steeper slopes than in reality. This is why other experiments with self-formed seepage channels were conducted with low-density sediment (Marra et al. 2014), and consequently the mobility ranged from 0.044-0.68.

We appreciate the reviewer pointing out some additional references that can be useful for us. We have incorporated them into the revised manuscript. We dispute the contention, however, that threshold sand channels in the lab are simply an issue of scaling. We also dispute that using plastic sediment allows one to make suspension rivers in the lab with non-cohesive sediment. A first important point we want to make is reporting Shields stress alone is not adequate to estimate transport stage for laboratory experiments. Lab experiments of channelized flows often span the laminar to turbulent

regimes, and the hydraulically-smooth to rough regimes; in addition, they have varying aspect rations that greatly affect flow resistance and roughness. In short,there is no way to accurately know what the critical shields stress is in a laboratory setting without a)explicitly measuring it, or b) allowing a channel to reach an equilibrium *and static geometry, without sediment feed*, and then inferring critical through a threshold channel assumption. The only effective way, in our opinion, to estimate critical given all of these issues is the second option; for the prescribed flow and grain size, allow a stable channel to evolve to a point of no transport by providing no sediment feed. This is what the IPGP group in France does (Seizelles et al., 2014-2014), and in transitional Reynolds number flows they get quite high critical Shields numbers, like 0.2 - 0.4. Note also that threshold Shields determined in laminar experiments can be quite high, up to 0.5 (Charru et al., JFM 2004; Lobkovsky et al., FGM 2008; many others). In the Marra et al. experiments, the high stress runs were indeed noted for the runs with lower density sediment, however the Reynolds numbers were also relatively low (72-541), potentially resulting in a higher than expected critical entrainment stress. Moreover, the river channels studied by Devauchelle et al. (2011) are not laboratory rivers; they are natural rivers that have both their beds and their banks comprised of non-cohesive sand. This system is indeed small and atypical, however in this case, the data show that this is a case of the exception proving the rule. In the rare case of a sand-bedded river with banks that are equally as erodible as the bed, we see that the system behaves identically to a gravel-bedded river with banks that are, at most, equally as difficult to erode as the bed. Finally, parallel experiments at UPenn and at Saint Anthony Falls Lab (U. Minnesota) attempted and failed to make self-formed suspension rivers by using low-density sediment and a range of discharges. These results haven't been published so we can't refer to them; but over multiple orders of magnitude, channels evolved so that the banks were at the threshold of motion.

Furthermore, there are a large number of other experiments with self-formed chan-

**ESurfD**
nels (compiled in Kleinhans et al. 2014 Fig. 9 and Table 1 and some more in 2015) that show a range of mobilities in the context of a large river dataset that also show a gradual transition from bedload to suspended load dominated behaviour. This includes high-mobility experiments with sand and floodplain-forming low-density sediment (Paola group, Minneapolis) and with materials that strengthen banks (famous experiments of Hoyal, ExxonMobil) (see Kleinhans et al. 2015 for references). The main point is that the nature of the bank material determines the channel geometry. This is essential for bar and river patterns and was incorporated in the hydraulic geometry in Kleinhans et al. (2015) because correct scaling of the bank material in experiments leads to desired channel aspect ratios. The qualitative understanding behind this work goes back at least to Ferguson (1987), and shows that the experimentally inferred relation of the straight-meandering-gravel transition as a function of slope by Schumm and Kahn (1972) is wrong and the bank stability is key. Furthermore, the natural and experimental channels adhere well enough to the hydraulic geometry relations of Parker et al. (2007) when accounted for the strength of the banks (Kleinhans et al. 2015), which allowed application of the Parker relation to bar theory. It is already well-known that the absolute dimensional shear stress in gravel-bed rivers is much larger than that in sand-bed rivers despite the lower sediment mobility, so that bank material unstable in the gravel reach can be stable in the sand-bed reach. The hypothesis on P10 L14-16 has already been voiced in a large body of bank erosion literature underlying the BSTEM model by Simon et al. (2000 and later references). This argument has not become more precise by the scatter plot in Fig. 7. Also the authors do not present new data or new insights; for example P10 L27-32 states that information about bank material is usually lacking and this is precisely why Kleinhans and van den Berg (2011) had to introduce that counterintuitive streampower measure and why Eaton and Church (2007) developed a rational regime theory that can be used to invert to bank stability measures from observed channel dimensions. These papers and many others already make that point; however, these references are lacking here.

We hope that the addition of our section summarizing regime theory, and describing

where our work fits in, clarifies the goal of our hypothesis. We have added even more references that we hope will rectify the perception that we are ignoring previous work (although we note that we were aware of most of this work already; it was left out only because we wanted to construct a simple and concise point without rehashing all of the work done on equibrium channel geometry, gravel-sand transitions, etc.) Nothing we say runs contrary to the work mentioned here; however, we strongly disagree that there is nothing new or more precise here. The issues raised above about bar theory are not relevant to the issue at hand. As for the other points; we do not dispute that these previous studies have addressed controls (or potential controls) of cohesion on bank geometry. But none of these papers have shown this: the Parker-type bedload model works to first-order for predicting channel geometry, for all rivers in which particles are large enough that cohesion doesn't matter. Data break away from this prediction only when bed material becomes small enough for cohesive bank materials (if present) to matter. We are not attempting here to invert each data point in the scatter to infer bank strength, or to introduce new parameters to model or explain things. The simple and novel point of this paper is that we can mark the transition from rivers not caring (to first order) about bank material, to caring about bank material; and that this happens because river sediments are bimodal AND that the smaller mode is the easiest-to-entrain material. We actually belive that this result is entirely consistent with all of the references raised by the reviewer; however, we stand firm that none of these previous references has explicitly made this point.

The final remark of the discussion is somewhat simplistic: "Some of the scatter in hydraulic geometry scaling plots may be due to stochastic fluctuations around the mean behavior." Here, stochastic is a rather dirty term for a number of well-known sources of variability. Apart from the fact that vegetation creates its own patterns with its own scales on the banks, leading to variations in channel dimensions, the balance between bank erosion and floodplain formation is not always in equilibrium because of

the specific length and time scales of bank failure and floodplain / levee formation. The assumption of the authors is that measured channel dimensions in their compiled dataset are due to such variations, but it is quite likely that the original authors of the datasets underlying this compilation already averaged out such variation. In fact, this is certainly the case for the Kleinhans and van den Berg (2011) dataset and in the experimental datasets from our laboratory, that were also taken up in the compilation of Metivier, and are therefore possibly part of the present compilation

This is a fair criticism. We have changed the text accordingly. Note that the main point here is not to explain variation within the two clouds of data (near-threshold and far above threshold), but rather to make a direct point about why there are two clouds. Reviewers are right to be interested in the origin and nature of the deviations from the first-order trend of the near-threshold channel. But this paper does not address that, and it cannot given the limited information from the datasets drawn upon. We have chosen to utilize a large dataset with minimal parameter information rather than a small dataset with more information — because we want to focus on the scaling and bi-modality.

**3   Reviewer 3**

On page 5, line 10, you state that you assume the Chezy friction factor is constant. I can see no possible justification for this assumption. Looking at Ferguson 2007 and Eaton and Church 2011, it is clear that Cf varies dramatically with relative roughness, and that changes in relative roughness also produce order of magnitude changes in the bedload sediment concentration generated at the same critical dimensionless shear stress. These scale variations are fundamental, and need to be included in the numerical analysis. This is one of the reasons that Millar took the numerical approximation approach that he did, and then generated power law equations using a Monte Carlo modelling approach (see Millar, 2005). This assumption appears to me to fundamentally undercut the entire following analysis.

Chezy friction factor can indeed vary dramatically with relative roughness, however it does not vary strongly with dimensionless discharge and has no discernible influence on sediment transport regime (see below figures). For these reasons, we do not believe that the assumption of a constant friction factor is damaging, or particularly relevant, to our analysis. More broadly, transport rate changes by orders of magnitude with small changes in fluid stress when one is near the threshold of motion. But, the differece between zero and a very small number can be many orders of magnitude, of course. What the reviwer is raising is related to the challenge of estimating critical. Flow resistance is one challenging piece to estimating critical, but it is not the only one. Indeed, the question of predicting critical remains a central challenge to geomorphology. But we do not believe that it will "fundamentall undercut the entire following analysis". First, related to the item above; while flow resistance varies quite a bit in our data, it is NOT bi-modal; so flow resistance cannot explain the apparent bi-modal transport states of alluvial rivers. Second — related to points made to the previous reviewer — this paper does not seek to explain the variance within clouds of data for sand and gravel rivers, but rather to explain the origin for two clouds instead of one. Finally, *we do not actually assume anything in the treatment of our hydraulic scaling data related to the friction factor.* Therefore, none of our analysis is undercut. Rather, we simply plot the predictions from the threshold "canal" theory as a reference point, following Metivier et al.'s work. It would be impossible to generate a threshold prediction for the global dataset using a site-specific and varying friction coefficient. You can view the straight line in the hydraulic geometry scaling plots as the simplest, dumbest model; constant Shields stress (which of course is also not true), constant friction, no cohesion, and no transport! We know that this is not a complete model for rivers; the deviations from this model tell us something. Friction varies among rivers but not systematically with discharge. Therefore, friction factor cannot explain the persistent offset in scaling between the threshold channel prediction and the data.

[Figure]

chezy.png

Page 1, Line 15: you claim that the empirical hydraulic geometry equation are robust, and show remarkably little variation in the exponents: this is a stretch. log log plots

chezy_rouse.png

hide the true scale of differences between datasets; the lumped data includes systems where W/d ratio declines downstream, and where it increases downstream; it includes systems where depth is nearly constant, and ones where it changes as expected. You also fail to observe that simple Froude scaling nearly explains the trends that are found in nature, which indicates that most of the variance can be explained simply by the size of the system, not any particular organizing principle. This is well described by Eaton in the Hydraulic Geometry chapter of the Treatise on Geomorphology. In fact, there are significant variations in the exponents, when they are compared against the Froude scaling exponents, and I believe this is well described by Eaton and Church (2007) in JGR

We do not deny that scatter exists for empirical hydraulic geometry equations; however we believe that given that the data show approximately one order of magnitude of scatter over approximately fourteen orders of magnitude(!) of range, the relationships can be described as robust. As for Froude scaling, we have two points. First, Froude scaling does not explain the bimodal nature of sediment transport states (see figure). Second, Hydraulic geometry relations (non-dimensionalized width, depth and slope as a function of discharge) are much tighter than any scaling with Froude number. This was already shown in Gary Parker's 2004 E-book, and can be shown with our data or any other; the most we can say about Froude number is that it increases with slope, doesn't exceed 1 by much, and shows a lot of scatter. Froude number and friction factor are similar in this regard; they show weak but detectable trends with some variables like slope; however, the variance around these trends is at least as large as the magnitude of the trends themselves. Finally, in the Treatise on Geomorphology, it is explicitly said that "An important drawback is that it is not possible in this [Froude scaling] framework to explicitly consider (or vary) the effect of bank strength on channel geometry; thus, one of the important independent variables is omitted." It is not clear what we would add by considering Froude scaling in this framework.

Page 1, Line 20: you attribute the concept of the dimensionless discharge to Metivier (2016). This is certainly not the primary source for a dimensionless discharge. Andrews used it in his 1984 paper (though there is an unfortunate typo in the final version, he used exactly this definition of a dimensionless discharge).

We thank the reviewer for this clarification and will cite Andrews as well. We also note of course that discharge has been non-dimensionalized by many including Parker's 1978 formulation. We are not trying to attribute the "concept" of dimensionless discharge to anyone in particular, since it's not important; rather, we are simply choosing to follow

froude.png

one kind of dimensionless discharge for comparison to recent work.

Page 2, Line 1 It is remarkable that you do not refer to Ferguson's classic work on the basis for regime theory. I believe that this is still one of the best summaries of the problem, and provides a much more satisfactory statement of the problem than the authors provide here

We thank the reviewer for this suggestion and have cited Ferguson. As mentioned in response to the editor's comments, our goal has never been to provide an exhaustive

review of all of the work that has been done on equilibrium channel geometry. This paper is about bimodal transport states and a potential explanation for the two modes.

Page 2, line 10: The "ground state" approach by Metivier sound exactly like the threshold channel approach used by many previous authors, including Lane, 1955; Henderson, 1966; Stevens, 1989.

We do cite Lane and Henderson. The term "ground state" was suggested as nomenclature to avoid the misconception that "equilibrium" implies a lack of dynamics. In other words, all channels wiggle around, and many colleagues we talked to were challenged by the formulation of the problem with threshold banks because they thought it was not consistent with dynamics. This is an attempt to clarify that point; that we are making a description of the river as if it is a static thing; in comparison to a natural river, you can think of this static description as describing the averaged behavior without reference to the scales of fluctuations around it.

Page 2, line 2: the discussion of the stable channel paradox seems to be to miss the point. The key thing to realize is that the shear stress acting on channel banks is lower than the average boundary shear stress, and that the shear stress acting on the bed is higher than average. This particular issue was very well addressed by Rob Millar's implementation of a regime theory that considered bank strength, using the shear stress partitioning approach by Knight, Flintham and Carling. I recommend reading Millar 2005 and the numerous relevant papers cited therein. I believe that Millar's contribution has adequately asked and answered the questions that this paper attempts to answer, and that the data analysis presented herein does not provide any additional insight to the problem. In any case, I do not see the justification for publishing this analysis without acknowledging the previous work!

Millar 2005 was focused on gravel-bedded rivers, not the transition from a bed-controlled regime to a bank-controlled regime. We hope that the inclusion of our paragraphs outlining regime theory and where are work fits in addresses this. Again, in the context of our paper, Millar's work is about addressing the variation of data within the gravel regime. Our paper is about why there are two clouds of data — and why one follows "regime theory" of any kind (Parker, Millar, etc.) and one does not. We're happy to cite Millar's work in our revised version as part of our new summary on regime theory. We just wish to re-emphasize that our paper is distinct from this; neither Millar or other workers showed that (near-)threshold theory predicts first order trends for all gravel rivers, but that rivers with smaller grains depart from this trend; and that the hinge point is related to where bed material becomes weaker than bank material. While we don't disagree that there are deviations from the threshold model for the gravel regime — which could be addressed by the work of people like Millar — we point out that the threshold model explains the first-order trend of all gravel rivers, while it definitely does not for sand bedded rivers.

page 3, line 11: you state that there is no accepted model for the equilibrium geometry of rivers far above threshold. I think is is an unfair and inaccurate representation of the current state of the science. There are models (like Millar and Quick, 1998) that successfully predict the geometry of such streams. They are published, and have been successfully tested, yet you appear to disregard all of these models without even bothering to mention them.

The data presented in Millar and Quick (1998) show gravel-bedded channels that have shields stresses up to approximately 3 times the threshold of motion. When we refer to rivers that are far above threshold, we are referring to the fine-grained rivers that are approximately 2 orders of magnitude in excess of the threshold of motion, for which

there has yet to be established a satisfactory understanding.

Analysis in Fig 2: does this really tell us anything that we do not already know from Church's (2006) exploration of the plotting positions of various rivers? While the plots are somewhat different, I do not see what novel insight they provide, particularly given the scale distortions introduced by the inappropriate assumption that Cf is constant!

We do not understand the reviewer's meaning of "plotting positions". Figure 2 was meant to demonstrate a)the robust nature of hydraulic scaling relationships, and b) that sand-bedded rivers with banks and beds composed of non-cohesive sand plot in the same geometric space occupied by gravel-bedded rivers. This is a context figure, not the fundamental result (which comes later), and we don't see why it shouldn't be included. Note that the plot itself *makes no assumption about Cf - so there is NO scale distortion!.* Only the lines showing the naive threshold model with constant everything would have such a "scale distortion." But note, as we pointed out above, that there is not positive scaling of friction factor with dimensionless discharge; and, therefore, friction cannot explain the deviation in scaling between the naive threshold model and the data.

Page 6, lines 1 to 5: You text is not an accurate representation of the bank strength issue. With respect to vegetation, the effect on relative bank strength is fundamentally scale dependent (see Eaton and Giles, 2009, Eaton and Millar, 2017), which you fail to mention (and which also introduces significant scale distortions), and the effect on channel with is close to a linear one. As banks become very erosion resistant, then changes in width are reduced, because the channel is able to reach the hydraulically optimal form, beyond with narrowing does not produce any increase in bed shear stress (basically this is the geometry predicted by Wobus 2004 for erosion into bedrock).

We do not dispute that vegetation can have an effect on bank strength. Our hypothesis does not run contrary to this line of thinking at all. The overarching idea of our hypothesis is that alluvial rivers are either bed controlled or bank controlled. To a first order, fine-grained rivers are bank-controlled, however I am certain that in some cases in gravel-bedded rivers, vegetation is strong enough to shift the control on channel geometry from the bed to the bank. Phillips and Jerolmack (2016) demonstrated that, to a first order, the geometry of gravel-bedded rivers can be explained by near-threshold channel geometry. This sort of model, is not design to describe variation around the trend. In this paper, we are demonstrating a systematic departure from that trend when the grain size on the bed gets small enough.

Page 6, line 9: your use of Schumm's M findings is a poor choice, since that analysis is a classic tautology. Clay and silt are never found on the channel bed, so Schumm's index (which is the percent silt and clay averaged over the entire channel boundary) builds in the width depth ratio; therefore it cannot be used to make a meaningful prediction of the width depth ratio. Simons and Albertson (1963) do manage to make some progress on the sedimentological controls for canals, however

We do not use Schumm's findings, we merely mention him as one of the first people to consider the influence of bank composition on channel geometry.

Page 6, line 15: the authors present a hypothesis about what controls bank erosion, but that hypothesis was advanced previously by Nanson and Hickin, and is the basis for the models developed by Millar and Quick (1993) and by Eaton (2006).

We hope that our paragraph summarizing regime theory provides clarification of the uniqueness of our hypothesis.

**4 Andrew Wickert**

p.1,l.17-19: I find Tal and Paola (2007) to be a key reference for meandering vs. braiding when involving vegetaion; perhaps it is just not cited due to the many others here, but I suggest that you read it if you haven't!

We do cite Tal and Paola 2007.

Equation 1: Consider splitting onto 3 rows for better readability

Done

p.2,l.1: remove indent

Done

Fig. 1: Why does this schematic have to have different bed and bank material? In particular, your statement that the center of the channel is only slightly in excess of critical Shields stress indicates that it is more likely all gravel that you are trying to illustrate. Do you intend to use this for both the gravel-bed and sand-bed channel examples? If so, just a little bit of clarification/generalization will be needed. (As a graphic, this is an excellent block illustration/generalization of the Parker (1978) line drawing, by the way, and one that I'd like to borrow for lectures.)

This is meant to illustrate a sand-bedded river. we have added clarification.

p.3,l.1: Parker (1978a) predicted... (no need for "Parker's model")

Done

p.3, l.22-24. Note also that Pfeiffer et al. (2017) demonstrate that gravel-bed rivers are significantly above the threshold of motion in rapidly-uplifting settings. They argue that this is due to armoring, though I have my own ideas about this (as-of-yet unpublished, and therefore nothing that you'll have to tangle with in a review). We, and others, have attempted to replicate the results of the Pfeiffer et al. paper with no success. Their results of their work are dependent upon the biases implicit in the assumption of slope-dependent critical shields stress. As their results cannot be replicated under more rigorous testing, we will not use that paper as a reference. See comments above to the Editor on this.

p.3,l.24: remove comma after "slope"

Done

Fig. 2: Show legend in only one panel to not overlap with data

Done

Fig. 3: It looks in the figure like you have one point from seepage channels and one from experiments. Therefore, I suggest that you reword "River channels ..." to take into account that this is not quite so plural, and is not tested to be generalizable.

The single, larger points are meant to illustrate the mean of multiple measurements taken along a single longitudinal profile or multiple runs in a laboratory setup.

p.8,l5-6: Which site-specific variations are you considering, and how/why are they important?

The term "site-specific" was meant to refer to the grain size and slope. We merely wanted to state that there are multiple ways of estimating critical.

p.11,l.1-3: Could you discuss other reasons for cohesion? Interlocking grains and capillary forces come to mind. Are these significant compared to the surface charge effects

Capillary forces would not be a factor because the bank toe is perennially saturated. We hypothesize that part of the reason for the huge range in bank critical shear stresses from previous work is because previous studies sampled from a variety of locations up the bank, which would then incorporate capillary forces despite those not having an effect at the bank toe. We do not believe that the interlocking of grains is a significant cohesive factor because grain interlocking would happen in both cohesive and non-cohesive settings, however we see that for non-cohesive systems rivers

adjust themselves to the threshold of motion. Perhaps grain interlocking is implicitly accounted for in this clustering around the threshold of motion.

p.11,l.3: material; –> material:

Done

p.11,l.31-32: Would you like to discuss some of the reasons for the low frequency of channels with 1-10 mm grain size? In particular, do you think that this may have to do with the crystal size / granule break-down problem, or possibly be connected to the transition between cohesion-dominated banks and particle-weight dominated banks that makes these grains either difficult to move or whisked away in a larger-clast gravel-bed river? This is of course ignoring arguments for equal mobility...

While this is indeed an interesting question, we do not believe the grain size gap found in channels is of particular relevance to this paper.

---

## Referee Report (RR1)

I have now carefully reviewed the responses of the other reviewers and the responses of the authors. I commend the other reviewers for their attention to the detailed history of the literature on grain-size transitions, while noting that my main focus was closer to the idea of bank control on the transition between gravel-bed and sand-bed rivers, which is what the authors have now clarified as their intent. I believe that the relative lack of criticism from my part was due to my realizing this focus. In hindsight, it may also be a direct result of having been investigating the same problem myself, and having come to the same general conclusion about the influence of the cohesive strength of muds on channel pattern, just a few weeks before I received this invitation to review. This primed my brain for the paper, and indeed contributed substantially to my enthusiasm for the work of the authors, but meant that my focus on the problem at hand caused me to not notice some of exposition required to set up the general background. Therefore, I am glad to have seen the more critical reviews of the other referees as well as the response of the authors and their revised manuscript.

I remain in agreement with the conclusion that around the sand–gravel transition, the width of rivers with sandy sediment is controlled by the cohesion of bank materials (this force being stronger than the weight of the particles), and those with gravel beds is controlled by the weight of the particles. I find this to be a simple and mechanistic line of reasoning with the ability to explain and unify a number of observations – many researchers have danced around this point, but this is the first time that I have seen it explicitly stated. I think that the work is worth publishing on the merits of this alone.

However, I do not agree with the authors' decision to forge ahead with their characterization of all gravel-bed rivers as threshold-state rivers while not noting that above-threshold gravel-bed rivers are a major conclusion of Millar and Quick (1998), and in fact refusing to cite the recent demonstration of this by Pfeiffer et al. (2017). In this, the authors' views are at odds with the evidence. I had thought that above-threshold gravel-bed rivers were simply omitted by accident the first time around, which lead to my earlier and more positive review; this omission could be easily addressed by citing the studies appropriately and noting that *most* gravel-bed rivers are near-threshold rivers. What I have trouble understanding is that the main addition of the submitted paper to the scientific knowledge, and the one that captured my enthusiasm (and continues to do so) is the gravel–sand transition. This makes this gravel-bed river threshold-state issue somewhat beside the point at hand, and one that unfortunately will cause me to withdraw my support for publication of this (in my opinion) otherwise-sound paper until it is addressed.

**Comments from the prior review that were not addressed by the authors:**

*Use* \citep{} *instead of* \citet{} *with extra parentheses.*

Please do this. Or maybe you were just using \cite{}? Whichever way it is, please just use BibTeX properly.

**Comments from prior reviews that were not satisfactorily addressed by the authors:**

*p.3, l.22-24. Note also that Pfeiffer et al. (2017) demonstrate that gravel-bed rivers are significantly above the threshold of motion in rapidly-uplifting settings...*

The authors state that they are unable to reproduce these results and therefore will not cite the paper. I do not find their arguments to be convincing for the following reasons:

1. The authors claim an issue with the statistical treatment used by Pfeiffer et al. without demonstrating what this is or how it might change the results (see response to AE Turowski on C2 and C3). Claiming an error on a piece of existing literature requires evidence.

2. The authors claim that the their conclusion is based on the biases involved in a slope-dependent $\tau_c^*$; I have tested their data set with both a constant and slope-variable (Lamb et al., 2008) $\tau_c^*$, and have obtained a range of $\tau_b^*/\tau_c^*$ from approximately 0.3 to 10 in both cases.

3. The authors argue that there was no independent measurement of $\tau_c^*$ in those rivers with high $\tau_c^*$: I have not checked this but $\tau_c^*$ typically changes over a factor of 2 wheras $\tau_b^*/\tau_c^*$ (at bankfull) ranges from $\approx 0.3$ to $\approx 10$. This therefore cannot explain the observations. Furthermore, to be fair to Pfeiffer et al. (2017), it is important to note that $\tau_c^*$ is often not measured independently in the field, making it difficult to do so with such a large compilation, and that for rivers in which $\tau_b^*/\tau_c^*$ is great, such observations could be difficult with available technology at best and dangerous at worst.

4. The authors incorrectly state that the authors do not provide data on hydraulic geometry. The supplementary inforamtion has full data on hydraulic geometry. The authors likewise state that the authors do not give information on discharge; while this is true, the data provided (slope, depth, width, grain size) can, with an appropriate roughness formulation, can be used to generate a reasonable estimate of discharge.

5. Finally, the authors claim that they were unable to reproduce the results of the Pfeiffer et al. study and failed to do so. However, it is unclear based on this response how the authors did this and/or what data sets they used (Pfeiffer's? Their own?) and whether the data set used for validation indeed was sufficient to test the ideas put forward in that paper. They claim that this means that the results cannot be replicated under "more rigorous testing". From my point of view, Pfeiffer et al. is a peer-reviewed paper with data that support the conclusion that gravel-bed rivers exist at above-threshold transport stage; the claimed "more rigorous testing" on the other hand, has no description to support it. The answer clearly must be to stick with Pfeiffer et al. until/unless future research provides an alternative explanation for their findings.

While I appreciate that a thorough analysis of the Pfeiffer et al. (2017) study may go beyond the scope of the paper, it seems that one is left with two choices. Either the authors should at least tacitly accept it and the idea that rivers may not be as tidy as they would like, or they must thoroughly disprove this. The former would require just a sentence noting that the are not looking at environments of rapid uplift, and therefore are not evaluating the explanation put

forward by Pfeiffer and co-authors. The latter would require a more extensive explanation. In my opinion, it is better to look at the publication of the two papers by Phillips and Jerolmack (2016) and Pfeiffer et al. (2017) as a challenge of how to look deeper into the data and validate whether or not the discrepancy exists, how to improve our measurement abilities, how to critically evaluate our assumptions, and (if necessary) how to unify theory.

*p.11,l.31-32: Would you like to discuss some of the reasons for the low frequency of channels with 1-10 mm grain size? In particular, do you think that this may have to do with the crystal size / granule break-down problem, or possibly be connected to the transition between cohesion-dominated banks and particle-weight dominated banks that makes these grains either difficult to move or whisked away in a larger-clast gravel-bed river? This is of course ignoring arguments for equal mobility...*

The authors state that this point is not important to this paper. However, there remains a fundemental question: does the bimodal transport state exist as the result of a bimodal input, a bimodal filter wihtin the system, or both? The authors seem to argue for the middle answer, but do not mention that it may instead be, for example, the result of a bimodal input to which the internal response of the system that the discuss herein is more a response and less a driver.

*In the authors' response to Reviewer 3 on the use of a constant Chézy friction factor, I have some open questions.*

First, how did they calculate the friction factor in the response? This is not stated.

Second, friction will change with the presence of bedforms. The authors explicitly do not address form drag. This is OK by me since they state this clearly. However, I question at this point whether a constant Chézy friction factor selection (0.1) is to look for deviations from a particular relationship, or if it is, as the authors state, that they simply do not care about reducing scatter. Could you please clarify this?

**Line-by-line comments:**

p. 1, line 15: (I did not know this during the first review round – sorry to bring it up only in round 2) Lacey was actually the first to generate a power-law relationship for hydraulic geometry. A nice review is by Savenije (2003): "The width of a bankfull channel; Lacey's formula explained".

p. 1, lines 20-21: I am not going to hold you to this because you are trying to generalize (as is appropriate here), but I will note that there is significant scatter in the power-law hydraulic geometry relationships, and would suggest that the scales of the scatter vs. the strengths of the relationships are appropriately acknowledged.

p. 2, line 5: in a rectangular channel (again, I missed this the first time around, but is easy to note)

p. 8, lines 27-29: Phillips and Jerolmack put significant effort into measuring $\tau_c^*$ and compiling measurements of it along gravel-bed rivers in the mostlytectonically-inactive mountain west of the USA and in Puerto Rico. (I know they included a river in the Oregon Coast Range, but study the low-relief portion of the river.) Their study excludes data from rapidly-uplifting landscapes required to address the Pfeiffer et al. (2017) argument as well as some of the references brought forth by Reviewer 3 on non-threshold behavior. Therefore, the authors have described only part of the sum of the knowledge, and have put forward a statement about threshold behavior based only on this. The cautionary note here is that it is easy to say "most rivers do this" but hard to say that "all rivers do this". The difference between "most" and "all" could hold some important information into the forcings and response. I appreciate that these may be beyond the scope of this paper, but I find it important to avoid such blanket statements and partial referencing of the literature on a particular problem that can result in a disjointed scientific literature.

p. 12, lines 1-2: The Kean & Smith reference does not support the statement about mud, vegetation, and erosion thresholds. (I missed this as well in round 1 because I had not yet read this paper, and therefore did not realize that it was mis-cited.)

p. 12, lines 32-33: Your paper is a really nice piece of work about the channel width transition between sand- and gravel-bed rivers. You do not address the question at all about all alluvial rivers being near-threshold (see above comments).

p. 14, lines 1-3: Such a statement may be true, and has been rephrased to be consient with Millar and Quick's work. However, without an evaluation of the Pfeiffer et al. work, it is not possible to say that that this is true in general.

p. 14, line 3: Contrary to my last comment, this statement is still at odds with Millar and Quick (although their range of $\tau_b^*/\tau_c^*$ is less than that reported by Pfeiffer et al.). I have gone back and checked this following Lamb et al. (2008), and my calculated $\tau_b^*/\tau_c^*$ values range from 0.6 to 2.6.

Conclusions (in general): it is possible that some material in the second half of your conclusions could go in your discussion.

---

## Referee Report (RR2)

**Reviewed manuscript**: Evidence of, and a Proposed Explanation for, Bimodal Transport States in Alluvial Rivers.

**Manuscript Authors**: Kieran Dunne and Douglas Jerolmack

**Journal**: Earth Surface Dynamics

**Reviewer:** Shawn Chartrand

General Comments:

The authors present a new hypothesis that all alluvial rivers tend to a state of near-threshold transport condition for the boundary material that is most difficult to mobilize. The bed material of gravel-bedded rivers conditions the near-threshold bankfull geometry. On the other hand, cohesive deposits located relatively low in the channel banks sets the geometry for sand-bedded rivers, when and where cohesive deposits occur. Figures 5 through 7 of the manuscript suggest that the new hypothesis has merit, and may offer a useful framework to help explain alluvial channel geometry from headwater to distal terminus. I think the paper will make a valuable contribution to literature.

The manuscript has undergone a revision following four earlier reviews. I have read through the previous comments and the responses. In many cases the authors adequately address the most pertinent comments. Some comments were not addressed, and in most cases the authors present a reasonable response, although it is clear that there is some disagreement about particular points. This is to be expected for new and big ideas. As a result, I think the revisions adequately address previous comments.

I have read the paper twice and recommend that the authors take an additional pass at improving the presentation of the material. I found the writing in places challenging to follow, and this unfortunately makes it more difficult to appreciate the novelty of the work. The challenging parts of the paper in terms of writing are focused within the middle part of the Introduction (the newly added paragraphs), and in the presentation of results. Figures are not adequately presented in Sections 3 through 5 and it is left to the reader to understand what the Figures present. Additionally, I found extra unnecessary words in some of the new sentences, a lack of needed legend material for Figure 2, and found Figure 1 hard to appreciate given how it is presently constructed. None of the suggested revisions below are difficult to address, but I encourage the authors to carefully step through the paper and revise the writing to be more clear, use less provocative or distracting terms, and use less parenthetical sentence structure. This style of writing made it difficult for me to follow your train of thought when it was used.  To assist the authors I offer suggestions for editorial revisions that may be helpful. If the suggestions misrepresent the science or message, that is a reflection of my misunderstanding only. Please do not be discouraged by my comments. I offer the comments because I think the work is valuable, and wish to see it presented in the best and most comprehendible way, for the benefit of the authors and the field.

Specific/Editorial Comments

Figures:

General comment that it was difficult to quickly determine grain size trends in your Figures related to sandy vs. gravelly beds. Without changing your color mapping, I think it would help readers if you used one symbol type for sandy beds and a different symbol type for gravel beds. The suggestion stems from

the fact that throughout your paper you use the words sandy and gravel, or derivatives of the words, and it sure would help in reviewing the Figures to have these sizes jump out.

Figure 1 – I had a very hard time seeing the cross-stream profile sketched in plane with the top of the figure. It would be much easier to see the profile if you rotated it up, and sketched it above the image from bank to bank. In this configuration it would sit perpendicular to the top plane of the graphic and inspection for the reader would be simple.

Figure 2 – I spent about ten minutes trying to figure out which symbols represent sand size sediments and which represent gravel. I could guess…then when I reached Figure 3 I saw your helpful colorbar for grain size. I suspect this is accidentally omitted from Figure 2? Why begin the panels with W*H; seems more natural to begin with W or H, then present W*H. This links to comments below. Last, is the light blue line a measure of error for the seepage data? If so it is not indicated in the caption.

Figure 3 – Second to last sentence in the caption is missing from other Figures. Either move this note to Figure 2 where it is first of relevance, or include wherever it is relevant.

Figure 4 – Caption calls out one point's color (cyan) but not the other (red). Keep it consistent.

Figure 5 – Panel D: how about color coding the bars to help readers relate to your other plots? Or place a vertical lines in the plot at 10^-3 and 10^-2 to highlight your separation of sand beds vs. gravel beds? The suggestion is focused on making it easier for the readers to link information between plots. You may want to indicate units for the x-axis in the Figure caption.

Figure 6 – This Figure is quite small in the PDF; I am not sure what the published size may be, but as presented it was hard to see the details you discuss. It is a nice presentation of Singer's (2010) data. Last sentence in your caption you use the phrase "far above threshold". Can you quantify or characterize this more precisely (i.e. an order of magnitude…)? As stated it is broad brushed and detracts from your work.

Manuscript:

Sections 3, 4 and 5 –

I read these sections several times and found most of the writing challenging to follow due to structure. There is one case where the authors actually present a Figure, what the axes are and what is plotted. The authors generally begin each section with a few prefatory sentences (or more) which in some cases provide a pre-summary of the results. Then results are presented and the section wrapped up. These three sections would greatly benefit from re-structuring of the writing. I encourage the authors to begin each section with presentation of the relevant Figures. Explaining what is plotted, etc. and then connecting their results with the other details they discuss. As it is presently structured, I interpret these three sections as more of a test of others ideas, rather than a presentation of their results, a test of their ideas, and then how it all fits in with the bigger picture. I realize the authors are avoiding the traditional Results section and that this is okay for previous reviewers and ESurf. So the level of discussion they provide seems justified. However, as written, the main results were a challenge to appreciate, and the lack of methodical review of each Figure detracts from the contribution. As a suggestion, begin Section 3 with " Figure 2 shows…". Then work through the results, weaving in Equation 7 where appropriate. The first sentence of the section is not needed.

Line 13: I read the sentence leading up to the ending a few times. I struggled with the last phrase. Tentative evidence seems tricky as a concept. It is more straightforward to just state that you present a set of results which supports your hypothesis or idea. Which certainly seems to be the case from my read of the paper.

Page 2 –

Line 1: $Q_*$ presents a huge range of parameter space, ranging over 14 orders of magnitude in Figure 2. From the formulation the range depends on how the flow magnitude compares to the $D_{50}$ raised to a power 5. For grain sizes from 0.0001 to 0.1 m, the square root of this term ranges in magnitude from approximately 1E-10 to 0.003. This may be more an observation, but it might be helpful to point this out because it is pretty uncommon to see a parameter space of 14 orders of magnitude in the associated literature. I can think of only a few, and they come from the same group as the authors.

Line 12: …"the addition of" is not necessary

Lines 19 – 27: These sentences are long and hard to follow. Edit for clarity. Here are some suggestions.

Line 22: …Suggest: "This line of thinking links with the second branch of regime theory established by Parker (1978a).  Parker (1978a) solved…"

Lines 26-27: Suggest: "…rivers that demonstrate that bedload dominated gravel-bedded rivers are slightly offset from a threshold channel…"

Page 3 –

Lines 1 – 15: These sentences are also long and hard to follow. Edit for clarity. Here are some suggestions.

Line 1: I don't understand the first part of the first sentence. Suggest: "The last branch of regime theory suggests that alluvial rivers optimize their geometry to maximize flow resistance and hence minimize the boundary fluid shear stress."

Lines 3 – 7: Break into two sentences.

Line 9: This is a particularly key sentence for your argument, and with respect to comments by Maartin Kleinhans. As written it is hard to understand. Suggest: "This paper highlights the bedload-transport state transition between gravel-bedded river segments explained by Parkers theory, and sand-bedded river segments which do not fit within Parkers theory."

Line 13: "…is the that…" the and that should be reversed.

Lines 16 - 18: Break into two sentences

Line 17: Naïve? This word distracts from your point.

Lines 24 - 29: I have read these sentences several times. Where exactly are these results presented? I reviewed both papers by Metivier et al. and it is not obvious to me how these sentences fit within the Metivier et al. papers. Please clarify.

Page 5 –

Lines 8 – 9: Second part of the first sentence is not needed.

Lines 24 – 26: Item (1) is hard to understand as written.

Page 6 –

Line 12: No indent needed. Is "simplicity" the best word? Seems like you used their values for comparison sake.

Page 7 –

Lines 2 – 4: There are other explanations beside a long timescale. Bed slope locally could adjust more readily (i.e. characterized by a shorter response time scale) than bank position, for example. Since you are plotting point values which reflect a range of length scales, my quick review of your data indicates you have a mix of length representations. I do not dispute the perspective of profile adjustment at the basin or many reach scale over relatively long times; but your data do solely reflect these conditions.

Page 8 –

Line 1: Naively? This word distracts from your point.

Lines 2 – 3: $Q\_*$ and $W*H$ are normalized by grain size. I don't understand your grain size point as a result.

Lines 8 – 9: The first sentence is confusing, and the second does not add much. Consider deleting both.

Line 23: Don't need conspicuous. The data position in the plot says it all.

Lines 23 – 24: Last sentence not needed. It only distracts from your message.

Line 31: I think you mean to reference Figure 5, not Figure 3.

Page 10 –

Line 1: I don't know where this result is presented.

Line 15: You discuss results which you do not present. Please show the results or delete the last sentence.

Page 11 –

Lines 15 – 20: Here are examples of how you use parenthetical structure to make many points at the same time. Please break the thoughts up and present the material in a manner that is easier to follow.

Page 12 –

Lines 1 – 5: Where is the material of the last sentence presented? I have no idea, but want to know.

Lines 7 – 22: I struggled with the main point of this paragraph. What is your main message and how does it link to the paragraphs around it? I could not piece it together.

Page 13 –

Line 2: There is less data in the 1-10 mm range to < or >, but does it really represent a paucity of data?

Lines 14 – 16: The last sentence doesn't really fit in with the paragraph. I think you need to link it better.

---

## Author Response (AR2)

**Wickert**

We thank the reviewer for his comments and suggestions. In an effort to diffuse debate around the precise characterization of gravel-bedded rivers as purely threshold channels, we have rephrased our conclusion to read, "We propose that all alluvial rivers, regardless of their bed material grain size, organize their hydraulic geometry such that they cluster around the threshold of motion for the most resistant material — the structural component of the channel that is most difficult to mobilize." We hope that by using the word "cluster" instead of "are", we do not come across as in disagreement with previous papers that have investigated order 1 deviations from the threshold of motion in gravel bedded rivers. We have included a sentence in the introduction that defines near threshold as an order 1 multiple of the threshold of motion and far-above threshold as an order 10-100 multiplier of the threshold of motion. We have furthermore included a sentence in the introduction that acknowledges previous work that has been done on order 1 deviation from the threshold of motion.

Use citep instead of citet with extra parentheses. Please do this. Or maybe you were just using cite? Whichever way it is, please just use BibTeX properly.

Thank you for the catch. Sorry about this persistent error

...The former would require just a sentence noting that the are not looking at environments of rapid uplift, and therefore are not evaluating the explanation put forward by Pfeiffer and co-authors. The latter would require a more extensive explanation. In my opinion, it is better to look at the publication of the two papers by Phillips and Jerolmack (2016) and Pfeiffer et al. (2017) as a challenge of how to look deeper into the data and validate whether or not the discrepancy exists, how to improve our measurement abilities, how to critically evaluate our assumptions, and (if necessary) how to unify theory.

We thank the reviewer for his suggestion and have included the following sentence acknowledging the variability around the threshold channel clustering: "Several studies have presented evidence that sediment supply and bank vegetation may drive gravel channels further above threshold (Pfeiffer et al.,2017; Millar and Quick, 1998) . Values for Shields stress in gravel-bed rivers reported for a wide range of environments, however, rarely exceed 2-3 times critical."

p.11,l.31-32: Would you like to discuss some of the reasons for the low frequency of channels with 1-10 mm grain size? In particular, do you think that this may have to do with the crystal size / granule break-down problem, or possibly be connected to the transition between cohesion-dominated banks and particle-weight dominated banks that makes these grains either difficult to move or whisked away in a larger-clast gravel-bed river? This is of course ignoring arguments for equal mobility...

The authors state that this point is not important to this paper. However,there remains a fundamental question: does the bimodal transport state exist as the result of a bimodal input, a bimodal filter within the system, or both? The authors seem to argue for the middle answer, but do not mention that it may instead be, for example, the result of a bimodal input to which the internal response of the system that the discuss herein is more a response and less a driver.

We thank the reviewer for reminding us of this previous criticism. There have been numerous papers that have been published on this subject and we don't believe, given the data and results presented in this paper, that we have much to add on this subject. We believe that we implicitly address this issue with figure 5 that compares the bimodality in shields stress to the bimodality in grain size.

In the authors' response to Reviewer 3 on the use of a constant Chezy friction factor, I have some open questions. First, how did they calculate the friction factor in the response? This is not stated. Second, friction will change with the presence of bedforms. The authors explicitly do not address form drag. This is OK by me since they state this clearly. However, I question at this point whether a constant Chezy friction factor selection (0.1) is to look for deviations from a particular relationship, or if it is, as the authors state, that they simply do not care about reducing scatter. Could you please clarify this?

We calculated the friction factor using $u = Cf\sqrt{H_{bf}S}$. This equation is a statement of how much of the body force acting on the fluid is translated into flow velocity. This is a consequence of form drag at all scales: grain, bed form, bar form, meanders, etc. In showing the friction factor plots in the previous review, we wanted to make the point that there is no strong trend with any other variable; therefore, changes in friction factor cannot explain the first-order hydraulic geometry scaling relationships. The only use of a constant friction factor is in the calculation of the threshold channel curve, which is only used as a reference line to examine systematic deviations from it. Chezy friction factor could explain some of the second-order variance in the data, but not the deviation for the trend lines for hydraulic geometry from the threshold prediction.

p. 1, line 15: (I did not know this during the first review round  sorry to bring it up only in round 2) Lacey was actually the first to generate a power-law relationship for hydraulic geometry. A nice review is by Savenije (2003): "The width of a bankfull channel; Lacey's formula explained". Thank you for the correction. We have corrected the opening sentence to read, "Almost 100 years ago, Lacey (1930) proposed an empirical relationship relating the width of an alluvial river to its water discharge. Leopold and Maddock (1953) built upon this to derive hydraulic scaling relations for bankfull channel geometry of

alluvial rivers."

p. 1, lines 20-21: I am not going to hold you to this because you are trying to generalize (as is appropriate here), but I will note that there is significant scatter in the power-law hydraulic geometry relationships, and would suggest that the scales of the scatter vs. the strengths of the relationships are appropriately acknowledged.

We thank the reviewer for his comment, and with all due respect to the reviewer, we believe that the graph clearly an approximately 1 order of magnitude of scatter over approximately 12 orders of magnitude range of data is a strong enough case to warrant the generalization that the hydraulic scaling trends are robust

p. 2, line 5: in a rectangular channel (again, I missed this the first time around, but is easy to note)

Noted, and has been included.

p. 8, lines 27-29: Phillips and Jerolmack put significant effort into measuring $\tau_{*c}$ and compiling measurements of it along gravel-bed rivers in the mostly tectonically-inactive mountain west of the USA and in Puerto Rico. (I know they included a river in the Oregon Coast Range, but study the low-relief portion of the river.) Their study excludes data from rapidly-uplifting landscapes required to address the Pfeiffer et al. (2017) argument as well as some of the references brought forth by Reviewer 3 on non-threshold behavior. Therefore, the authors have described only part of the sum of the knowledge, and have put forward a statement about threshold behavior based only on this. The cautionary note here is that it is easy to say "most rivers do this" but hard to say that "all rivers do this". The difference between "most" and "all" could hold some important information into the forcings and response. I appreciate that these may be beyond the scope of this paper, but I find it important to avoid such blanket statements and partial referencing of the literature on a particular problem that can result in a disjointed scientific literature.

With all due respect to the reviewer, this comment is most pertinent to Phillips and Jerolmack in which order 1 magnitude deviations from the threshold of motion discussed. This current paper is only about the separation of near-threshold (order 1 multiple of threshold) channels and far above threshold (order 10-100 multiple of threshold) channels. The results of this paper do not lean on Phillips and Jerolmack, nor are these order 1 deviations from threshold pertinent to the results presented in this paper. For clarification to readers, we have modified the following sentence to define what we mean by "near threshold" and "far-above threshold": "We address these questions by re-analysis of existing data. We revisit the global data compilations of Li et al. and Trampush et al.and argue that natural rivers appear to exhibit bi-modal transport states

corresponding to near threshold (order 1 multiplier of threshold) and far-above threshold (order 10-100 multiplier of threshold)."

 The Kean and Smith reference does not support the statement about mud, vegetation, and erosion thresholds. (I missed this as well in round 1 because I had not yet read this paper, and therefore did not realize that it was mis-cited.)

Thank you for the catch and we apologize for this error. We have changed the sentence and have included more appropriate citations: "We do not consider vegetation explicitly; however, we note that numerous studies have analyzed the effects of vegetation on erosion thresholds (Micheli and Kirchner, 2002; Abernethy and Rutherfurd, 2001).

 Your paper is a really nice piece of work about the channel width transition between sand- and gravel-bed rivers. You do not address the question at all about all alluvial rivers being near-threshold (see above comments)

Sorry, we don't quite understand the reviewer's criticism. The data from Singer (2010) was used in this paper to illustrate the the bimodality that is present in the global datasets is also reflected in an individual river. The focus of this paper has always been the bimodality that exists across all alluvial rivers and an attempt to identify attractors of these two modes.

 Contrary to my last comment, this statement is still at odds with Millar and Quick (although their range of $\tau_{*b}/\tau_{*c}$ is less than that reported by Pfeiffer et al.). I have gone back and checked this following Lamb et al. (2008), and my calculated $\tau_{*b}/\tau_{*c}$ values range from 0.6 to 2.6.

We acknowledge the reviewer's point, however we must point out that for every point noted to be above the threshold of motion, there is one below the threshold of motion. This is why in the paper we use the term "cluster" to describe the threshold of motion as an attractor. We have included a sentence in the paper to define near-threshold as an order 1 multiplier of threshold and far above threshold to be an order 10-100 multiplier of threshold.

Conclusions (in general): it is possible that some material in the second half of your conclusions could go in your discussion.

We feel that the material in the conclusion is necessary because we want to present the threshold-limiting material hypothesis and acknowledge that there is still a lot of work to be done to validate or invalidate it. If the reviewer believes that the conclusion is too long, we can remove the sentence "Consideration of the slope- or grain-size-dependence of the critical Shields stress shows that alluvial rivers are bi-modal in terms of transport stage and bed-material grain

size, and that these modes correspond generally (but not always) to bed-load gravel rivers and suspension sand rivers." as this is covered in the Discussion section.

**Chartrand**

We thank the reviewer for his comments and suggestions and we have made our best effort to address them.

General comment that it was difficult to quickly determine grain size trends in your Figures related to sandy vs. gravelly beds. Without changing your color mapping, I think it would help readers if you used one symbol type for sandy beds and a different symbol type for gravel beds. The suggestion stems from the fact that throughout your paper you use the words sandy and gravel, or derivatives of the words, and it sure would help in reviewing the Figures to have these sizes jump out

We thank the reviewer for his suggestion, however we believe that it would be better practice to treat the data as a continuum and allow patterns of distinction to emerge organically.

Figure 1 – I had a very hard time seeing the cross-stream profile sketched in plane with the top of the figure. It would be much easier to see the profile if you rotated it up, and sketched it above the image from bank to bank. In this configuration it would sit perpendicular to the top plane of the graphic and inspection for the reader would be simple.

The angled perspective was chosen to be able to showcase the lateral change in the stress profile.

Figure 2 – I spent about ten minutes trying to figure out which symbols represent sand size sediments and which represent gravel. I could guess. . . then when I reached Figure 3 I saw your helpful colorbar for grain size. I suspect this is accidentally omitted from Figure 2? Why begin the panels with W*H; seems more natural to begin with W or H, then present W*H. This links to comments below. Last, is the light blue line a measure of error for the seepage data? If so it is not indicated in the caption.

We thank the reviewer for this catch and apologize for the confusion. There was a mixup uploading the figures to the latex document. The colorbar has been added and a statement of the meaning of the cyan line.

Figure 3 – Second to last sentence in the caption is missing from other Figures. Either move this note to Figure 2 where it is first of relevance, or include wherever it is relevant.

Thank you for the suggestion. We have moved it to the caption of figure 2.

Figure 4 – Caption calls out one point's color (cyan) but not the other (red). Keep it consistent.

Thank you for the catch, we have made the correction.

Figure 5 – Panel D: how about color coding the bars to help readers relate to your other plots? Or place a vertical lines in the plot at $10^{-3}$ and $10^{-2}$ to highlight your separation of sand beds vs. gravel beds? The suggestion is focused on making it easier for the readers to link information between plots. You may want to indicate units for the x-axis in the Figure caption.

Thank you for the suggestions. We have changed the labels on the x-axis to be more consistent with the other plots. We would prefer not to colorcode each bar as we think having a big colorful graph would detract from the goal of this figure which is to have a simple histograph that clearly shows a bimodal distribution to the reader, which might not be as apparent in the scatter graphs presented earlier.

Figure 6 – This Figure is quite small in the PDF; I am not sure what the published size may be, but as presented it was hard to see the details you discuss. It is a nice presentation of Singer's (2010) data. Last sentence in your caption you use the phrase "far above threshold". Can you quantify or characterize this more precisely (i.e. an order of magnitude. . . )? As stated it is broad brushed and detracts from your work.

Thank you for the suggestion. In response to both reviewers, we have now modified a sentence in introduction to explain that by near threshold, we mean an order 1 deviation from the threshold of motion, and by far-above threshold, we mean an order 10-100 deviation from the threshold of motion. Sorry, don't know what happened regarding the size of the image. We have enlarged it.

As a suggestion, begin Section 3 with " Figure 2 shows. . . ". Then work through the results, weaving in Equation 7 where appropriate. The first sentence of the section is not needed.

With all due respect to the reviewer, we believe that our way better links the broader idea of regime theory and hydraulic geometry to the data.

Page 1 - Line 13: I read the sentence leading up to the ending a few times. I struggled with the last phrase. Tentative evidence seems tricky as a concept.

It is more straightforward to just state that you present a set of results which supports your hypothesis or idea. Which certainly seems to be the case from my read of the paper.

Thank you for the suggestion, however we believe that "tentative evidence", while admittedly not the strongest of phrases, is the best descriptor of what this paper provides. The work leading up to this paper has lead us to the threshold-limiting material hypothesis, the validity of which is indicated by the data presented in this paper but not directly supported. Thus, we believe that the use of the phrase "tentative evidence" is most appropriate.

Page 2 – Line 1: $Q_*$ presents a huge range of parameter space, ranging over 14 orders of magnitude in Figure 2. From the formulation the range depends on how the flow magnitude compares to the $D_50$ raised to a power 5. For grain sizes from 0.0001 to 0.1 m, the square root of this term ranges in magnitude from approximately $1E - 10$ to 0.003. This may be more an observation, but it might be helpful to point this out because it is pretty uncommon to see a parameter space of 14 orders of magnitude in the associated literature. I can think of only a few, and they come from the same group as the authors.

Thank you for the suggestion, however in dimensional space, the range is already 10 orders of magnitude, and given that we present the equation that non-dimensionalizes discharge by grain size before presenting this graph, we believe that the large range of the data is explained.

Line 12: . . . "the addition of" is not necessary
Removed

Lines 19 – 27: These sentences are long and hard to follow. Edit for clarity. Here are some suggestions. Line 22: . . . Suggest: "This line of thinking links with the second branch of regime theory established by Parker (1978a). Parker (1978a) solved. . ." Lines 26-27: Suggest: ". . . rivers that demonstrate that bedload dominated gravel-bedded rivers are slightly offset from a threshold channel. . ."

Thank you. We have made the suggested changes.

Page 3 – Lines 1 – 15: These sentences are also long and hard to follow. Edit for clarity. Here are some suggestions. Line 1: I don't understand the first part of the first sentence. Suggest: "The last branch of regime theory suggests that alluvial rivers optimize their geometry to maximize flow resistance and hence minimize the boundary fluid shear stress."

We mean to explain that there are multiple types of "regime theory" that exist at the moment and are practiced with little to no interaction with one

another. That is why the word "parallel" was chosen.

 Break into two sentences.
Done

 This is a particularly key sentence for your argument, and with respect to comments by Maartin Kleinhans. As written it is hard to understand. Suggest: "This paper highlights the bedload-transport state transition between gravel-bedded river segments explained by Parkers theory, and sand-bedded river segments which do not fit within Parkers theory."

With all due respect to the reviewer, we would prefer to leave the sentence as is because Parker's theory is shown to be applicable to rivers with bed and banks made up of uniform, non-cohesive material, regardless of grain size.

 ". . . is the that. . . " the and that should be reversed.
Thanks for the catch.

 Break into two sentences
With all due respect to the reviewer, we believe that, given that this is one idea, we believe it is best kept as a single sentence.

 Naïve? This word distracts from your point.
We have removed it.

 I have read these sentences several times. Where exactly are these results presented? I reviewed both papers by Metivier et al. and it is not obvious to me how these sentences fit within the Metivier et al. papers. Please clarify.

We just mean to say that gravel-bedded rivers follow the threshold prediction with a slight offset and sand-bedded rivers follow threshold prediction with a larger offset

 Second part of the first sentence is not needed.
We have removed the second half of that sentence.

 Item (1) is hard to understand as written.
With all due respect to the reviewer, we believe this wording is the best way

to get the point across.

 No indent needed. Is "simplicity" the best word? Seems like you used their values for comparison sake.

We have removed the indent. We would prefer to keep the term simplicity, because we are not doing a comparison between our results and those presented by the IPGP group, rather we are just using their framework to present data in this paper.

 There are other explanations beside a long timescale. Bed slope locally could adjust more readily (i.e. characterized by a shorter response time scale) than bank position, for example. Since you are plotting point values which reflect a range of length scales, my quick review of your data indicates you have a mix of length representations. I do not dispute the perspective of profile adjustment at the basin or many reach scale over relatively long times; but your data do solely reflect these conditions.

We acknowledge the possibility that locally this can happen. However, it has been reported from field data and models of alluvial river profiles that regrading the slope of an alluvial river happens slower. Generally, everybody reports scatter in slopes and we are just referring to the scatter that is reported.

 Naively? This word distracts from your point.
We have removed the word "naively"

 $Q_*$ and $W * H$ are normalized by grain size. I don't understand your grain size point as a result.

What we mean is that there is no offset from the central trend for fine particles in the upper range, as we see in plots of dimensionless width and depth.

 The first sentence is confusing, and the second does not add much. Consider deleting both.

With all due respect to the reviewer, we believe it is important to explain why we look at shields stress instead of just geometry to understand the transport condition.

 Don't need conspicuous. The data position in the plot says it all.

We have removed the word "conspicuous".

 Last sentence not needed. It only distracts from your message.
We have removed this sentence.

Line 31: I think you mean to reference Figure 5, not Figure 3. The bimodality is evident in both figures 3 and 5.

Page 10 – Line 1: I don't know where this result is presented.
Thanks for the catch. We have removed the sentence

Line 15: You discuss results which you do not present. Please show the results or delete the last sentence.
The lack of bimodality in slope or depth is implicit in the relatively consistent fluid shear stress across the GST shown in Figure 6.

Page 11 – Lines 15 – 20: Here are examples of how you use parenthetical structure to make many points at the same time. Please break the thoughts up and present the material in a manner that is easier to follow.
With all due respect to the reviewer, each of these sentences makes exactly 1 point each. We don't know how we can make it easier to follow.

Page 12 – Lines 1 – 5: Where is the material of the last sentence presented? I have no idea, but want to know.
van Rijn (2016) as referenced in equation 6.

Lines 7 – 22: I struggled with the main point of this paragraph. What is your main message and how does it link to the paragraphs around it? I could not piece it together.
This paragraph builds off of the previous one that states that the shields stress is not a relevant parameter for cohesive materials. This paragraph explains why and provides examples of critical shear stresses for cohesive substrates.

Page 13 – Line 2: There is less data in the 1-10 mm range to ¡ or ¿, but does it really represent a paucity of data?
"Paucity" to us is only meant as the dictionary definition. We are agnostic

as to the origins, however that has been looked into by a number of people.

Lines 14 – 16: The last sentence doesn't really fit in with the paragraph. I think you need to link it better

With all due respect to the reviewer, the purpose of that last sentence is to relate the broad topic of the paragraph (dynamics vs. statics) to the data presented in this paper. In the paragraph, we discuss how there are central trends in the face of large variability that become apparent when enough data is able to average out the noise. By presenting potential sources of that noise, we hope to give readers a better understanding of our differentiation between signal and noise.

[revised manuscript text omitted]